# Tracing winter temperatures over the last two millennia using a NE Atlantic coastal record

Irina Polovodova Asteman[1, 2], Helena L. Filipsson[3], Kjell Nordberg[1]

5  [1] Department of Marine Sciences, University of Gothenburg, Carl Skottsbergsgata 22B, SE 41319 Gothenburg, Sweden

[2] Currently at: Marin Mätteknik (MMT) Sweden AB, Sven Källfelts gata 11, SE 42671, Gothenburg, Sweden

[3] Department of Geology, University of Lund, Sölvegatan 12, SE 22362 Lund, Sweden

*Correspondence to*: Kjell Nordberg (kjell.nordberg@marine.gu.se)

**Abstract.** We present 2500 years of reconstructed bottom-water temperatures (BWT) by using a fjord sediment archive from the NE Atlantic region. The BWT represent winter conditions due to the fjord hydrography and associated timing and frequency of bottom-water renewals. The study is based on a ca. 8-m long sediment core from Gullmar Fjord (Sweden), dated by [210]Pb and AMS [14]C and analysed for stable oxygen isotopes ($\delta^{18}O$) measured on shallow infaunal benthic foraminiferal species *Cassidulina laevigata* d'Orbigny 1826. The BWT, calculated by using the palaeotemperature equation of McCorkle et al (1997), range between 2.7 - 7.8°C and are within the annual temperature variability, instrumentally recorded in the deep fjord basin since the 1890s. The record demonstrates a warming during the Roman Warm Period (~350 BCE – 450 CE), variable BWT during the Dark Ages (~450 – 850 CE), positive BWT anomalies during the Viking Age/Medieval Climate Anomaly (~850 – 1350 CE) and a long-term cooling with distinct multidecadal variability during the Little Ice Age (~1350 – 1850 CE). The fjord BWT record also picks up the contemporary warming of the 20[th] century (presented here until 1996), which does not stand out in the 2500-year perspective and is of the same magnitude as the Roman Warm Period and the Medieval Climate Anomaly.

## 1 Introduction

The climate variability over last two millennia has been widely recognized as crucial for the understanding of the present and future climate responses to anthropogenic forcing (e.g. Cunningham et al., 2013; Pages2k, 2013; McGregor et al., 2015; Abram et al., 2016). To evaluate how significant regional climate changes are or if observed temperature anomalies are unprecedented in view of long-term climate evolution, there is a need for long historical instrumental climate records. A major challenge for the reconstructions of past climate changes, both by using proxy data and paleoclimate modelling, is often a lack of such long instrumental records, which if available seldom reach beyond the 20[th] century. The North Atlantic region plays in this respect a paramount role for climate variability and global carbon budget by modulating the Atlantic Meridional Overturning Circulation (AMOC) (e.g. Eiríksson et al., 2006; Lund et al., 2006; Park and Latif, 2008; Trouet et al., 2009). The upper northern limb of the AMOC, the North Atlantic Current (Fig. 1A), delivers heat, salt, and nutrients from the tropics to the mid- and high latitudes and carries major parts of the volume flux into the Nordic Seas (Hansen and Østerhus, 2000). The AMOC is thought to be linked to the Atlantic multidecadal oscillation (AMO; Enfield et al., 2001) through sea surface temperature variability and it is connected to decadal variability of the North Atlantic Oscillation, (NAO; Curry and McCartney, 2001), where the NAO index is defined as the normalized sea level pressure difference between the Icelandic Low and the Azores High (Hurrell et al., 1995). The variability of AMOC also contributes to a multidecadal modulation of El Niño-Southern Oscillation (ENSO) (Ortega et al., 2012 and references therein). In addition, the North Atlantic Current passes between the subpolar and subtropical gyres (Fig. 1A), from which it draws water and, hence, depends on variability occurring within both gyres (Hansen and Østerhus, 2000). Variability of ocean temperature in high latitude North Atlantic and Nordic Seas are reflected in NW European climate and in winter Arctic sea ice extent (Årthun et al., 2017). Model projections predict that the AMOC will slowdown in response to future warming and enhanced Arctic

freshwater fluxes (e.g. Schmittner et al., 2005; Ortega et al., 2012; Caesar et al., 2018) with potential impacts on climate, ecosystems, agriculture and economy of many European countries (e.g. Kuhlbrodt et al., 2009; Jackson et al., 2015, Knox et al., 2016). Hence, high-resolution paleoceanographic records, which preferably overlap with instrumental observations and historical data, are needed from the eastern North Atlantic region in order to document climate variability related to physical properties of the North Atlantic Current and AMOC strength. At the same time many of the marine records available from the region to date tend to have low temporal resolution due to their location in the deep-sea or within the open shelf areas. Sediment archives of temperate fjord inlets located within the eastern North Atlantic region offer the potential of high-resolution records of maritime climate, because they act as sediment traps resolving climate variability at nearly annual resolution (Howe et al., 2010). Yet to date, there are relatively few such high-resolution paleoclimate records from the eastern North Atlantic fjords spanning the late Holocene (among others Mikaelsen et al., 2001; Klitgaard-Kristensen et al., 2004; Cage and Austin, 2010; Filipsson and Nordberg, 2010; Hald et al., 2011, Kjennbakken et al., 2011; Faust et al., 2016).

Meanwhile, crucial knowledge has been gained from temperature proxy datasets available from the North Atlantic and northern hemisphere in general, which represent either composite records of different climate characteristics with various temporal resolution or are a combination of historical and proxy data; with generated data sets mostly reflecting summer conditions at higher latitudes (e.g. Moberg et al. 2005; Gunnarsson et al., 2011; Butler et al., 2013; Cunningham et al., 2013; PAGES2K, 2013, 2017; Sicre et al., 2014; Linderholm et al., 2015). In contrast, based on instrumental records, increased winter temperatures have been suggested as an important driver of the most recent warming (Cage and Austin, 2010) and, hence, climate proxies incorporating winter signal are needed.

Herein, we present a bottom water temperature proxy record from the Gullmar Fjord, on the west coast of Sweden, which illustrates the climate development in NW Europe in over the last ~2500 years. Among advantages of the presented record are its high temporal (annual to sub-decadal) resolution and that a winter temperature signal is recorded in fjord foraminiferal shells (tests), due to specific hydrographic conditions. The reconstructed temperatures are based on stable oxygen isotopes ($\delta^{18}O$) measured in tests of a shallow infaunal foraminifer *Cassidulina laevigata* d'Orbigny 1826 and reflect the deep-water temperatures in the fjord basin. The fjord has a >100-yr long record of instrumental observations from the deepest basin, performed since 1869 (Fig. 2A-C); furthermore, >100-yr long time series of air temperature observations are also available for Stockholm, Sweden and central England. These instrumental observations of bottom water - and air temperatures are used to evaluate the accuracy of the reconstructed climate variability for the last century provided by the fjord sediment archive.

## 2 Study area

Gullmar Fjord is a Skagerrak fjord inlet, which is 28 km long and 1-2 km wide and oriented south-west to north-east (Fig. 1). The maximum basin depth is 118.6 m. The fjord is located at critical latitude picking up fluctuations between cold and temperate climates and has almost no tidal activity. The adjacent Skagerrak largely determines the local hydrography so that

the deep (basin) water, which is typically exchanged in the fjord during the winter, originates from the North Sea surface water flowing into the Skagerrak with the present-day current circulation system (Svansson, 1975; Nordberg, 1991). The 42-m deep sill at the fjord entrance restricts the water exchange and results in water column stratified due to salinity differences (Fig.1C). At the surface (<1m) there is a thin layer of river water from the Örekilsälven (Fig.1), which does not significantly impact the fjord hydrography (SMHI, 1994; Arneborg, 2004). Below, at 1-15 m water depth, there is a brackish water mass (S=24-27), primarily derived from the brackish Baltic current flowing northward along the Swedish west coast. The brackish water mass has a residence time of 20-38 days in Gullmar Fjord (Arneborg et al., 2004). A more saline water mass (S=32-33) at ~15-50 m is derived from the Skagerrak and has mean residence time of 29-62 days (Arneborg et al., 2004). The last and deepest layer (>50 m), referred herein as deep water or basin water, is more stagnant, with little seasonal and inter-annual changes in salinity ranging between 34 and 35 and inter-annual temperature variability of 4 - 8°C (Fig. 2A, B). The deep water temperatures vary between the years depending on the temperature of the inflowing water mass but remain stable seasonally (Fig. 2D). The deep-water salinities seasonally do not vary much from the average value of 34.5 (Fig. 2B). The stratification of the water column is further strengthened during the summer by the development of a strong thermocline, which impedes deep-water exchange. The deep-water exchange of the fjord basin water takes place once a year during winter, mostly between January and March, which is determined by using long-term instrumental records from the fjord (Arneborg et al., 2004). Due to a presence of a sill, isolating the fjord deep-water from the adjacent sea, and the comparably large basin volume, the winter temperature and salinity of the inflowing North Sea/Skagerrak water, are "annually preserved" in the fjord basin until the next deep-water turnover the following year (Arneborg et al., 2004). This results in a deep-water environment characterised by winter temperatures. The benthic foraminifers reproduce and grow in the fjord during the spring and summer (Gustafsson and Nordberg 2001), thus incorporating this annually preserved winter temperature signal of the ambient deep-water into their shells. This results in a stable oxygen isotope signal mainly reflecting winter temperatures of the North Sea surface water and the Skagerrak intermediate water flowing into the fjord during deep-water exchange.

The deep-water exchange in the fjord is driven by wind forcing, and largely depends on wind direction and wind strength (Björk and Nordberg, 2003). The latter two properties, in turn, are governed by the NAO, which is the dominant mode of climate variability in the region during the winter (Hurrell, 1995). In Gullmar Fjord, the higher frequency and duration of NE winds, common during the negative NAO index periods, result in Ekman transport of surface water from the coast and facilitate coastal upwelling, which causes the deep-water exchange (Björk and Nordberg, 2003). In contrast, a positive NAO index causes prevailing westerly winds, which limit the chances of the deep-water renewals to occur. From the late 1970s the NAO has been in its prolonged positive phase and is believed to be one of the triggers of severe seasonal hypoxia (<1 ml $O_2$ $l^{-1}$) in the deep fjord basin (Nordberg et al., 2000; Björk and Nordberg, 2003; Filipsson and Nordberg, 2004).

After an extensive deep-water exchange event in the fjord the oxygen level starts to decline in June, and the lowest oxygen levels normally develop between November and January, indicating hypoxic conditions (<2 ml $O_2$ $l^{-1}$), but so far anoxia has not been recorded (Fig. 2F). The first ever documented severe hypoxic event was noted in February 1890 by

Pettersson and Ekman (1891). In the following, severe hypoxic events were measured in 1906, 1961/62, and 1973/74 (Fig. 2C) but due to the low observation frequency and duration of these events are not well documented. Since 1979, multiple episodes of more frequent severe hypoxia lasting for at least 3 months have been observed. These events occurred in 1979/80, 1983/84, 1987/88, 1988/89, 1990/91, 1994/95, 1996–1998, 2008, 2014/2015, 2016 (e.g. Filipsson and Nordberg

2004a; Polovodova Asteman and Nordberg 2013; SMHI SHARK-database, 2017; Nordberg et al., unpubl. data).

The severe hypoxia makes the fjord basin hostile for large burrowing organisms but allows benthic meiofaunas to thrive. This lowers sediment bioturbation and results in well-preserved environmental sediment archive. The fjord basin has high sediment accumulation rates, which provide a high temporal resolution corresponding to 1-6 years per 1-cm thick sediment sample. Finally, the fjord sediment archive is characterized by the diverse and abundant foraminiferal faunas and

dinoflagellate cysts, which have already provided some insights in climate evolution and associated environmental changes on the Swedish west coast during the last two millennia (Filipsson & Nordberg, 2004a; Harland et al., 2006; Nordberg et al. 2009; Filipsson and Nordberg 2010; Polovodova et al., 2011; Harland et al., 2013; Polovodova Asteman & Nordberg, 2013; Polovodova Asteman et al., 2013).

## 3 Material and Methods

This study is based on a composite record of two sediment cores: GA113-2Aa and 9004, which were both collected at 116 m water depth at the same site in the deepest Gullmar Fjord basin (58°17.570' N, 11°23.060' E) (Fig. 1), for which the long-term hydrographic observations are available (Fig. 2A-C). The core 9004 (731-cm long) was taken with a gravity corer (Ø=7.6 cm) onboard *R/V Svanic* in July 1990. The core GA113-2Aa (60-cm long) with an intact sediment-bottom water interface was recovered by using a Gemini corer (Ø=8 cm) in June 1999 from the *R/V Skagerak*. In the laboratory both cores

were split in two halves and sectioned in 1-cm intervals. One half was used for bulk sediment geochemistry (TC, TN and C/N ratio), stable oxygen and carbon isotopes, dinoflagellate cysts- and benthic foraminiferal faunal analyses. Another half was stored as an archive at the Department of Geosciences, University of Gothenburg. The TC and stable carbon isotope data from both cores are published in Filipsson and Nordberg (2010), dinoflagellate cysts data are discussed in Harland et al. (2006, 2013), while C/N and foraminiferal assemblage data are presented in Filipsson & Nordberg (2004a), Polovodova et

al. (2011) and Polovodova Asteman et al. (2013). We also present data from the gravity core G113-091, collected at the same location as GA113-2Aa & 9004 onboard *R/V Skagerak* in September 2009, and used herein only (similar to our previous study) to create a composite age model for the cores GA113-2Aa and 9004 (Polovodova Asteman et al., 2013; see below).

In addition to the above-mentioned cores, we also use six surface samples (0-1cm) collected at five stations in the

Skagerrak (OS4, OS6, OS14, 9202 and 9205) and one station in the Gullmar Fjord (G113-091a: the same location as for

GA113-2Aa & 9004) in 1992-93 and 2009, respectively (Fig.1B, C; Table 1). All surface samples were stained by rose Bengal to distinguish individuals presumably living at the moment of sampling from the empty foraminiferal shells.

## 3.1 Sediment core dating and age model

The age model for the composite GA113-2Aa - 9004 record has been previously published in Filipsson and Nordberg (2010) with further revisions by Polovodova et al. (2011) and Polovodova Asteman et al. (2013). Eleven intact marine bivalve shells were recovered in life position from the core 9004 and were subject to the AMS $^{14}$C analysis (Fig. 3A; Table 2). All $^{14}$C dates were obtained through analysis at the Ångström Laboratory (Uppsala University, Sweden) and originally were calibrated using the marine calibration curve (Reimer et al., 2004; Bronk Ramsey, 2005). Ages were normalized to δ$^{13}$C of − 25‰ according to Stuiver and Polach (1977), and a correction corresponding to δ$^{13}$C= 0‰ (not measured) versus PDB has been applied. Herein we present ages recalibrated by using Calib Radiocarbon Calibration software v. 7.1 (Stuiver et al., 2017: http://calib.org/calib/), the most recent marine calibration curve (Reimer et al, 2013) and a reservoir age of 500 yr (ΔR=100±50), which has been obtained on pre-bomb marine bivalve shells from the Gullmar Fjord, provided by the Natural History Museums in Gothenburg and Stockholm (Nordberg and Posnert, unpubl. data). All ages are presented as median probability with 1-σ error margin (Table 2). Two dates at 98 cm and 313 cm showed minor age reversals and were omitted from the final age model (Table 2). The core GA113-2Aa was dated by using $^{210}$Pb and a constant rate of supply (CRS) model (Appleby and Oldfield, 1978), which suggested that the core material was deposited between ca. 1915 and 1999 (Fig. 3A). For details regarding GA113-2Aa age model see Filipsson and Nordberg (2004a).

Together the cores GA113-2Aa & 9004 proved to be a continuous sediment record with no gap in between based on correlation of the stable carbon isotopes (δ$^{13}$C) and benthic foraminiferal species *C. laevigata, Adercotryma glomerata* (Brady, 1878) and *Hyalinea balthica* (Schröter in Gmelin, 1791) with respective data from the core G113-091 (Fig. 3B herein; Polovodova Asteman et al., 2013; Polovodova Asteman and Nordberg. 2013). The composite record of GA113-2Aa & 9004 spans from approximately 350 BCE to 1999 CE (Table 2, Fig. 3A), and includes the late Holocene climate events such as the Roman Warm Period (RWP: ~350 BCE – 450 CE), the Dark Ages Cold Period (DA: ~450 – 850 CE), the Viking Age/Medieval Climate Anomaly (VA/MCA: ~850 – 1350 CE), the Little Ice Age (LIA: ~1350 – 1850 CE) and the contemporary warming from 1850 CE to present (Lamb, 1995; Filipsson and Nordberg, 2010; Harland et al., 2013; Polovodova Asteman et al., 2013; Helama et al., 2017). We add the Viking Age to the Medieval Climate Anomaly following the approach of Filipsson and Nordberg (2010), based on historical evidence that warming in Northern Europe began earlier than 1000 CE, which allowed Vikings to reach the NE coast of England and loot the monastery of Lindisfarne in 793 CE (Morris, 1985). For further details on chronology of the cores GA113-2Aa and 9004 see Filipsson and Nordberg (2004a), Polovodova et al., 2011; and Polovodova Asteman et al. (2013).

Combining the long gravity core with the 60 cm long Gemini core, which includes the sediment-bottom water interface and, hence, the intact core top, resulted in a high-resolution temporal record of almost 1-year cm$^{-1}$ sample for the upper part

of the record and <10 years cm$^{-1}$ sample for the deepest part of the record. Calculations from the [210]Pb analyses and the AMS-[14]C dates suggest sediment accumulation rates of ~9 mm year$^{-1}$ in the most recent sediments and approximately ~2.8 mm year$^{-1}$ in the compacted deepest part of the gravity core (Fig. 2). Hence, due to high accumulation rates the upper 60 cm of the record can be directly compared to instrumental hydrographic - and meteorological data (Figs 6, 7).

## 3.2 Stable oxygen isotopes

We measured δ[18]O on tests of shallow infaunal foraminifer *Cassidulina laevigata* from the core top samples and from the ca. 8-m long G113-2Aa - 9004 record (Fig.1B). Between 12 and 20 specimens of *Cassidulina laevigata* were picked from each sample for the analysis. In total 6 and 425 samples were analysed for stable oxygen isotopic composition for the surface sediments and composite G113-2Aa - 9004 record, respectively. All samples were measured at the Department of Geosciences, University of Bremen, Germany, using a Finnigan Mat 251 mass spectrometer equipped with an automatic carbonate preparation device. Isotope composition is given in the usual $δ$-notation and is calibrated to Vienna Pee Dee Belemnite (V-PDB) standard. The analytical standard deviation is <0.07‰ for δ[18]O based on the long-term standard deviation of an internal standard (Solnhofen limestone).

The temperature was reconstructed using the salinity: δ[18]O$_w$ relationship established by Fröhlich et al. (1988) (eq. 1), which is representative for this region (Filipsson, unpubl. data). An average salinity value of 34.4 (range 33-35) was used in equation 1, based on instrumental measurements between 1896 and 1999 for the fjord deep-water (station Alsbäck Deep). The salinity ($S$) was assumed to be constant over the investigated time period.

$$\delta^{18}O_w = 0.272 \times S - 8.91 \tag{1}$$

To calculate temperatures the paleotemperature equation by McCorkle et al. (1997) was applied (eq. 2). This equation is more appropriate to the temperature range observed in temperate fjord basin than the more commonly used linear equation by Shackleton (1974), which produces unrealistically high temperatures in our study (see results section). The bottom water temperature in degrees Kelvin ($T$ °K) was calculated as follows:

$$T^oK = \sqrt{\frac{2.78*10^3}{\ln(\frac{\delta^{18}O_c+1000}{0.97006*\delta^{18}O_w-29.94}+1000)}+\frac{2.89}{10^3}} \tag{2}$$

Here, $\delta^{18}O_c$ stands for stable oxygen isotopic ratio [18]O/[16]O measured in calcite tests of *C. laevigata*, while $\delta^{18}O_w$ is the isotopic composition of water calculated from equation 1 and converted from SMOW to V-PDB by subtracting 0.27‰ (Bemis et al., 1998)

Finally, to convert reconstructed temperatures to degrees Celsius, equation 3 was used:

$T°C=T°K-273.15$ (3)

Since 1990 *C. laevigata* has become a rare species in the Gullmar Fjord deep basin (Fig. 6), which resulted in a short gap in the most recent part of the record (see discussion). Similar gaps in $\delta^{18}O$ and, hence, in bottom water temperature data are also seen for the earlier part of the record and are due to absence or very low abundances of *C. laevigata* (Fig. 6).

**3.3 Hydrographical and meteorological instrumental data**

Long-term hydrographical instrumental data for temperature, salinity and dissolved oxygen concentration [$O_2$] for the fjord basin (average for 110-118 m w.d.) were extracted from the Swedish Meteorological and Hydrological Institute (SMHI) SHARK database (https://www.smhi.se/klimatdata/oceanografi/havsmiljodata/marina-miljoovervakningsdata). Some of the Gullmar instrumental data is also available from the Water Quality Association of the Bohus Coast (BVVF) (http://www.bvvf.se/), while the data prior to 1958 come from Engström (1970). The Skagerrak hydrography data for the stations adjacent to OS4-6, 9202, 9205 and OS14 were obtained from the International Council for the Exploration of the Seas (ICES: http://www.ices.dk/marine-data/).

Meteorological observations of air temperature were also obtained for Stockholm (https://www.smhi.se/klimatdata) and the Central England (http://www.metoffice.gov.uk/), which both have the longest historical meteorological records going as far back as the 18th century.

**4 Results**

**4.1 Core tops**

To obtain an error estimate and to facilitate the choice of the paleotemperature equation we used living (stained) tests of *Cassidulina laevigata* from the core top samples collected in the Gullmar Fjord and the adjacent Skagerrak. Calculated bottom water temperatures based on the $\delta^{18}O_c$ values from the living (stained) *C. laevigata* were compared to ICES and SMHI hydrography data from the adjacent stations (Fig. 4A). Also the $\delta^{18}O_c$ values predicted from the chosen equation (see below) were used to estimate the reliability of our temperature reconstruction (Fig. 4B). *Cassidulina laevigata* has been previously suggested to calcify 0.19‰ lower than equilibrium (Poole et al., 1994). Our $\delta^{18}O_c$ data from the core tops demonstrate an offset, ranging between 0.01‰ and 0.27‰ (mean 0.15‰), compared with $\delta^{18}O_c$ predicted using the palaeotemperature equation from McCorkle et al (1997) (Fig. 4B). Applying the mean correction of +0.15‰ to the Gullmar $\delta^{18}O_c$ record results in bottom water temperatures ~0.5-1°C higher than those recorded by instrumental observations in the fjord (Fig. 2A), while uncorrected $\delta^{18}O_c$ values produce temperatures close to observations. Taking the latter into the account and because, based on available data, it is difficult to estimate how large the correction should be, we further report the

uncorrected $\delta^{18}O_c$ values both for the core tops and for the sediment cores. Instead, we use a median value (0.7°C) of the range in produced temperature offset (Fig. 4A) as an error margin for our paleotemperature reconstructions (Figs 5-6).

Instrumental temperature data from ICES and SMHI were used to calculate $\delta^{18}O_c - \delta^{18}O_w$ for the core top samples to facilitate the choice of a paleotemperature equation. Plotting $\delta^{18}O_c$-$\delta^{18}O_w$ versus observed temperature data for different paleotemperature equations (Fig. 4C) allows estimating which of the equations gives the best possible agreement with the core top data and, hence, is the most appropriate for temperature reconstructions. Figure 4C shows that $\delta^{18}O$ values from the NW Skagerrak (OS4 and OS6) are clearly in better agreement with equations by Hays and Grossman (1991) and McCorkle et al. (1997), while the central Skagerrak samples (9202 and 9205) plot close to the linear equation by Shackleton (1974). The samples from Gullmar Fjord (G113-091) and the OS14 station, collected just outside the fjord, occupy a space in between the Shackleton equation and those by Hays and Grossman (1991) and McCorkle et al. (1997). This suggests that applying the Shackleton equation for Gullmar Fjord and Skagerrak will result in temperatures higher than observations, which has been also observed for *Cibicidoides* and *Planulina* from Florida Straits (Marchitto et al., 2014). Indeed, when testing the Shackleton equation on our dataset, the temperatures are warmer than the ICES hydrographic observation data by 1.5-2°C. In contrast, the equation by Bemis et al. (1998) applied to the core top $\delta^{18}O_c$ data produces the coldest temperatures, which are 0.9-1.9°C colder than observations. In turn, it appears that by using Hays and Grossman (1991) or McCorkle et al. (1997) equations, the corresponding calculated temperatures come closer to observations. Both equations are nearly identical for the temperature range 3-8°C (Fig. 4C) observed between 1890 and 2001 (Fig. 2) and by exercising both equations on Gullmar Fjord $\delta^{18}O_c$ record the almost identical paleotemperature curves are produced. This is rather curious since the equation of Hays and Grossman (1991) is based on meteoric calcite of non-biogenic origin. For this reason, in the current paper we apply the McCorkle et al (1997) equation for the paleotemperature reconstructions.

### 4.2 Composite record of G113-2Aa and 9004 sediment cores

The $\delta^{18}O$ record from the Gullmar Fjord shows both decadal and centennial variability for the last 2500 yr (Fig. 5A) and can be divided into five major isotopic intervals: 1) For the lower part of the record at 802–592 cm, corresponding to ~350 BCE – 450 CE, the $\delta^{18}O_c$ values are generally lower (~2.4‰) than the long-term average of 2.7‰. 2) Between 598 and 475 cm (~ 425 – 900 CE) the $\delta^{18}O$ record demonstrates a considerable variability (Fig. 5A), starting with higher $\delta^{18}O_c$ (2.8-3‰) at 598–574 cm (~ 425 – 525 CE), which then become lower (~2.4‰) at 574–529 cm (~ 525 – 700 CE) and increase again (~3.0‰) between 529 and 497 cm (~ 700 – 825 CE). 3) The 475–302 cm interval (~ 900 – 1350 CE) displays again lower $\delta^{18}O_c$ (~2.4-2.5‰), which are below the long-term average. 4) From 302 to 53.5 cm (~ 1350 – 1900 CE) the stable oxygen isotope record increases again with the majority of the $\delta^{18}O_c$ values being ~3.1–3.2‰ and exceeding the long-term average. Within this interval the highest $\delta^{18}O$ values of >3.2‰ are found between 300 and 170 cm (~1350CE – 1580 CE). 5) Finally, the $\delta^{18}O$ record becomes lower again (~2.4‰) between 53.5 and 5 cm (~ 1900 and 1996 CE). The $\delta^{18}O$ data for the

samples between 5 and 0 cm (1996-1999) are missing because we did not find enough specimens of *Cassidulina laevigata* to perform isotopic analyses.

Shifts of ~0.25‰ in $\delta^{18}O_c$ occur throughout the Gullmar Fjord $\delta^{18}O$ record, which according to the equation of McCorkle et al. (1997) used herein may potentially indicate a temperature variability of ~1°C. A corresponding salinity change is rather
small (0.02), calculated using the mixing line by Fröhlich et al. (1988) and by applying the $\delta^{18}O_c$ range of 2.6-2.85 and a corresponding temperature range of 4.9-5.9°C. Such salinity changes are well within the amplitude of inter-annual variability (1-1.5), recorded by instrumental salinity observations since the 1890 (Fig. 2B). Foraminifera precipitate their tests during several months (e.g. Filipsson et al., 2004) and thus integrate the inter-monthly salinity signal, which together with annual variability is minimal according to the instrumental data. For the upper part of the record 1-cm sediment slice integrates one
or possibly two growing seasons of *C. laevigata* and, hence, records a potentially higher variability of both salinity and temperature. In the deepest part of the record, however, a single 1-cm sample may correspond to ~7-10 years and, thus, more likely averages inter-annual salinity and temperature variability providing "a more smoothed" signal.

Stable carbon isotopes ($\delta^{13}C$) data from the composite G113-2Aa – 9004 record (Filipsson and Nordberg, 2010) were plotted against the oxygen isotope data presented herein, to investigate the potential relationship between the two e.g. due to
different water masses (Suppl. Fig. 1). No such relationship was found (Suppl. Fig. 1), which indicates that our $\delta^{18}O$ record mainly reflects fjord deep-water temperatures.

### 4.3 Reconstructed bottom water temperatures (BWT)

The resulting calculated bottom water temperature record is plotted both as absolute temperature values (Fig. 5B) and as anomaly from the mean value (5.4°C), based on the instrumental temperatures observed between 1961 and 1999 (Fig. 6).
With very few outliers, the reconstructed temperature range (2.7 - 7.8°C) is within the present-day annual variability, documented from instrumental temperature measurements in the fjord deepest basin since 1890 (Fig. 2A-C; Fig. 5C). To further prove that our record represents a winter signal rather than summer conditions (as most biological proxies) we compare the obtained BWT record to instrumental temperatures recorded in the fjord deep water during summer and winter. When performing such a comparison, instead of the commonly used June -August (JJA) temperatures for designation of
meteorological summer, one has to consider the observations during May-August, when foraminifera precipitate their calcite (Gustafsson and Nordberg, 2001; Filipsson et al., 2004), which is used herein for stable isotope analysis. Likewise, instead of months used for definition of meteorological winter (December-February: DJF), when comparing our record to instrumental data we use January-March, which define "hydrographic winter" in the fjord and are associated with months when deep-water exchanges occur (see Study area).
Observed annual temperatures registered between 1890 and 1996 (which corresponds to the uppermost part of the composite G113-2Aa – 9004 record) vary between 3.0 and 8.3°C, which gives an amplitude of 5.3°C. Corresponding instrumental 1890-1996 temperatures for foraminiferal growth season in the fjord (May-August: see above) show a 4.1 -

7.2°C range with an amplitude of 5.4°C. When studying the reconstructed temperatures over the last 2500 years the corresponding amplitude, i.e. the difference between the maximum (7.8°C) and the minimum (2.7°C) temperatures is 5.1°C (Fig. 5B). Also when plotting the reconstructed bottom water temperatures for the period 1890 –1996 versus corresponding instrumental bottom water temperatures as annual average and means for May-August (Fig. 7B) and January-March (Fig. 7C), the calculated bottom water temperatures and hydrographic data agree with each other rather well in terms of amplitude. An increased agreement, however, is reached when comparing the reconstructed data to the hydrographic winter (January-March) temperature (Fig. 7C), which is not surprising considering the fjord hydrography and a season when deep-water exchanges typically occur (see Study Area section). Hence, Gullmar Fjord $\delta^{18}$O-based temperature record reflects the winter temperature variability of surface water in the North Sea.

From the reconstructed Gullmar Fjord temperature record five bottom water temperature intervals can be recognized (Figs 5B, 6), in parallel to the isotopic intervals mentioned above. 1) From ~350 BCE to 450 CE the fjord bottom water temperatures are consistently above 5.4°C, the 1961-1990 mean. 2) Between 450 CE and 850 CE the record fluctuates between positive temperature anomalies (~450–650 CE) and negative anomalies (~650 – 850 CE) reaching minimum value at ~750 CE. 3) At ~850–1300 CE the bottom water temperatures are again above the average with a short negative anomaly around 1200-1250CE. 4) The period between ~1300 CE and 1850 CE in the Gullmar Fjord record is unprecedentedly cold for the last ~2500 years with the majority of temperature anomalies being negative and reaching the minimum value around ~1350 CE (Fig. 6). 5) Finally, from ~1850 CE towards present day the record is characterised by consistently positive bottom water temperature anomalies, which are comparable in the amplitude to the high anomalies found at ~350 BCE – 450 CE.

**4.4 Gaps in the record due to absent/rare *Cassidulina laevigata***

Some intervals in the G113-2Aa - 9004 record were barren of *C. laevigata* tests and hence for those intervals $\delta^{18}$O values and the corresponding reconstructed bottom water temperature data are missing. Those intervals are: ~130-120 BCE, ~725-740 CE, ~1260-1265 CE, ~1273-1277 CE, ~1340 CE and ~1996-1999 (Fig. 6). The most recent period of absent/rare *C. laevigata* in the Gullmar Fjord coincides with higher bottom water temperatures and frequently occurring severe hypoxia (see introduction and discussion), as registered by the instrumental measurements in the fjord deepest basin (Fig. 2C).

**5 Discussion**

The Gullmar Fjord winter bottom water temperature record shows both centennial and multidecadal variability and has a striking resemblance with climate periods (see below) historically known in the northern Europe over the last 2500 years (e.g. Lamb 1995; Stuiver et al., 1995; Moberg et al., 2005; Filipsson and Nordberg, 2010; Helama et al., 2017). The record demonstrates periods of temperature variability, which correspond to the Roman Warm Period (~350 BCE – 450 CE), the

Dark Ages cold period (~450 – 850 CE), the warm Viking Age/Medieval Climate Anomaly (~850 – 1350 CE), the colder Little Ice Age (~1350 – 1850 CE) as well as the warmer conditions during the 20$^{th}$ century (~1850 CE –present). There is an overall cooling trend in the Gullmar Fjord temperature record for the last 2500 years, which is consistent with other climate proxy records for this period (e.g. Lebreiro et al., 2006; Eiriksson et al., 2006; Hald et al., 2011, McGregor et al., 2015).

Among forcing mechanisms for the late Holocene climate variability in the North Atlantic region changes in temperature and influx of the Atlantic Water to the region (e.g. Nordberg, 1991; Hass, 1996; Klitgaard-Kristensen et al., 2004; Eiriksson et al., 2006; Lund et al., 2006), radiative forcing (Jiang et al., 2005; Hald et al., 2007), volcanic activity (Otterå et al., 2010; McGregor et al., 2015), land-use changes and increased greenhouse gas emissions (e.g. Masson-Delmotte et al., 2013; Abram et al., 2016) are suggested. In addition, there is a strong coupling between atmospheric - and ocean circulation linked

to the NAO variability. The NAO influences strength and frequency of moist westerly winds, bringing precipitation to the Northern Europe and has even been suggested to induce multidecadal-scale changes in the AMOC (Dickson et al., 1996), which on centennial scales are linked to the late Holocene major climate extremes (Bianchi and McCave, 1999). Below we discuss each of the climate extremes in detail and compare our record to available temperature proxy data from other settings, highly influenced by the multidecadal NAO variability and climate changes associated with it.

**5.1 The Roman Warm Period (prior to ~ 450 CE)**

The fjord record shows consistently positive bottom water temperature anomalies during the Roman Warm Period (RWP) when compared to 5.4°C, the annual mean for 1961-1999 (Fig. 6). The RWP is often associated with increasingly warm and dry summers both on the British Isles and in the central Europe and is linked to the expansion of the Roman Empire (Lamb, 1995; Wang et al., 2012). The RWP warming coincided with a more vigorous flow of the Iceland Scotland Overflow Water,

which is an important component of the AMOC modulating the European climate (Bianchi and McCave, 1999). Other studies report an increased contribution of the Atlantic water to the East Greenland shelf, a reduced sea ice concentration and an increased export of fresh water from the Arctic with the East Greenland Current (Fig. 1A), which all are thought to be linked to a shift from the negative to a positive NAO after ~500 BCE/0 CE and changes in the AMO regime (e.g. Perner et al., 2015 and references therein; Kolling et al., 2017). Harland et al (2013) analysed dinoflagellate cysts from the same

composite core as presented herein and, based on observed changes in species composition, suggested that sea surface temperatures (SSTs) in the fjord were >10°C during the RWP, as compared to the present-day SSTs of ~9°C (SMHI, 2017). Other studies suggest SSTs of 6-10°C for the waters off N Iceland (Sicre et al., 2011), 10.7-12.6°C for the Vøring Plateau, Norwegian Sea (Risebrobakken et al., 2011), >13°C for off the NW Scotland (Wang et al., 2012) and >15°C for the Rockall Trough, NE Atlantic (Richter et al., 2009) during this period. Also for the coastal NW Atlantic (Chesapeake Bay) the SSTs

as high as 12-15°C were reported (Cronin et al., 2003).

For the adjacent Skagerrak an increase in both intermediate and bottom water temperatures is reported based on Mg/Ca data on benthic foraminiferal species *Melonis barleeanus* (Butruille et al., 2017). The authors demonstrate a ~2°C

temperature increase and report a temperature range of ~6-8°C during the RWP. In a 2000-yr long temperature record from the Malangen Fjord, NW Norway (Hald et al., 2011), the RWP is characterized as "a warm period with stable bottom water temperatures". The Malangen fjord record is based on δ¹⁸O measured on *Cassidulina neoteretis* Seidenkrantz 1995 and documents a bottom water temperature range of 5.5-7.5°C (Hald et al., 2011). Both Skagerrak and Malangen Fjord studies agree well with our dataset, which demonstrates a temperature increase of ~2.5°C, resulting in a 5.4-7.9°C temperature range during the RWP for the Gullmar Fjord deep water (Fig. 5). The somewhat higher upper range limit of the RWP bottom water temperatures in the Skagerrak and Malangen Fjord, compared to our data, may be explained by a more direct influence of the more temperate Atlantic water at those sites, which may be less prominent in our study area as it is more land-locked and with a stronger continental influence. Also given that our record reflects winter temperatures, its lower BWT temperature range during the RWP is quite reasonable.

When comparing our data to the major temperature synthesis efforts done for the last two millennia, it becomes evident that our RWP reconstruction seem to disagree with the northern hemisphere temperature record of Moberg et al (2005), which is mostly characterized by the negative RWP temperature anomalies (Fig. 6). On the other hand, the warming seen in the Gullmar Fjord dataset is consistent with the PAGES2k temperature synthesis for the continental Europe (Fig. 6), which also reports a distinct warming corresponding to ~2-3°C temperature increase during the RWP (PAGES2k, 2013).

**5.2 The Dark Ages Cold Period (~ 450 – 850 CE)**

Our record displays variable bottom water temperatures in the fjord during the Dark Ages (Figs 5-6), which is initiated with a short-living negative anomaly at ~ 400-450 CE, then switches to positive values (~450-650 CE) and then becomes negative again at ~650-850 CE. The Dark Ages Cold Period (DACP) is commonly linked to a large-scale human migration in the central Europe (Lamb, 1995; Büntgen et al., 2011). The DACP was contemporaneous with a reduced flow of the Iceland Scotland Overflow Water (Bianchi and McCave, 1999), low solar activity, low pollen influx (Desprat et al., 2003), glacier advance (Lamb, 1995) and a negative mode of the NAO (e.g. Seidenkrantz et al., 2007; Orme et al., 2015; Helama et al., 2017). Summer temperatures <10°C in French Alps (Millet et al., 2009), increased humidity in the northern Europe (Barber et al., 2004) as well as a widespread abandonment of arable lands and cultivation in the SW Norway (Salvesen, 1979) were also documented for this period. Seidenkrantz et al. (2007) also report a warming of subsurface waters off West Greenland during the DACP attributed to a stronger Atlantic component of the West Greenland Current and a negative NAO.

There is also some cooling during the DACP indicated for the intermediate and deep water in the adjacent Skagerrak (Butruille et al., 2017) but lower temporal resolution makes it difficult to directly compare the Skagerrak record with ours. In contrast, variable SSTs during the Dark Ages are reported by some North Atlantic records (Sicre et al., 2011; Risebrobakken et al., 2011), with timing similar to the variability of the Gullmar Fjord temperatures (see above). Variable bottom water temperatures are also reported for the Malangen Fjord with a range (5.5-7.5°C) relatively close to our results (~4-8°C). There is also some fluctuation between cooling and warming with a ~3-4°C amplitude in a Mg/Ca –based SST record from the

Chesapeake Bay (Cronin et al., 2003), as well as in the DACP temperatures reconstructed for the continental Europe (PAGES2k, 2013).

## 5.3 The Viking Age / Medieval Climate Anomaly (~ 850 – 1350 CE)

After the Dark Ages the bottom water temperature anomalies in Gullmar Fjord become positive between ~ 850 CE and 1350 CE, which fits well with the onset of the warming during the VA/MCA. The warm MCA is believed to be associated with a positive NAO index (e.g. Trouet et al., 2009; Faust et al., 2016), which likely have strengthened the AMOC (Bianchi & McCave, 1999) and resulted in an increased transport of heat and moisture to the higher latitudes. The MCA also coincided with Grand Solar Maximum at 1100–1250 CE (Zicheng and Ito, 2000) and its temperature optimum occurred between 1000 CE and 1300 CE when there was a sharp temperature maximum in most of Europe (Lamb, 1995).

The mean annual northern hemispheric and continental Europe temperature records (Moberg et al., 2005; PAGES2k, 2013) show the onset of warming as early as between ~850 and 950 CE, with distinct warmth peaks reached around 1000 CE and 1100 CE and the MCA termination around 1300 CE, which all agrees with our data rather well (Fig. 6). The Malangen Fjord record also shows the warming already before 800 CE, which terminates around 1250 CE (Hald et al., 2011), a century earlier than in the Gullmar Fjord record. Despite such inconsistency in timing, which likely results from dating uncertainties (which may be the case for both studies), the two fjord records agree with each other rather well in terms of reconstructed bottom water temperature ranges for this period: 5.4-7.6°C for the Gullmar Fjord and 5.5-7.1°C for the Malangen Fjord. In the adjacent Skagerrak both intermediate and deep-water temperatures are reported to increase from ~6 to 8°C (Butruille et al., 2017) but sampling resolution of the former is too low for the MCA period. In turn, bottom water temperatures in the Scottish Loch Sunart also increased by ~1.2°C during the MCA (Cage and Austin, 2010), which is also within the abovementioned ranges. An increase of similar magnitude during the MCA is also reported for the sea surface temperatures in the North Atlantic (Cunningham et al., 2013).

An interesting feature in the Gullmar Fjord record of the VA/MCA is a presence of a short-lived cooling centred at ~1250 CE before the final peak of warmth at 1250-1350 CE (Fig. 6: see blue box). Such short cooling during the MCA is also documented for both eastern and western Atlantic coasts (Chesapeake Bay: Cronin et al., 2003; Loch Sunart: Cage and Austin, 2010) but with a slightly different timing, either due to dating uncertainties or application of different temperature proxies (Mg/Ca vs $\delta^{18}$O).

## 5.4 The Little Ice Age (~ 1350 – 1850 CE)

From ~1350 CE to ~1850 CE our record shows winter bottom water temperatures 2-3°C lower than the instrumental annual mean for 1961-1999 (Fig. 6). Many other proxy records report cooling of similar magnitude or even stronger in the North Atlantic during the LIA (e.g. Stuiver et al., 1995; Cronin et al., 2003; Klitgaard Kristensen et al., 2004; Eiríksson et al., 2006;

Hald et al., 2011; Sicre et al., 2011). The PAGES2k synthesis of marine palaeoclimate records spanning the past 2000 years also identified a robust global surface ocean cooling with the coldest conditions from 1400 to 1800 CE (McGregor et al., 2015). The Little Ice Age is commonly associated with glacier advances in the Arctic and alpine regions (Porter, 1986; Miller et al., 2012) in response to reduced solar activity (Mauquoy et al., 2002) and summer insolation (Wanner et al., 2011),

increased volcanism (Miller et al., 2012), negative North Atlantic Oscillation (e.g. Trouet et al., 2009; Faust et al., 2016) and reduced strength of the AMOC (e.g. Bianchi & McCave, 1999; Klitgaard Kristensen et al., 2004; Lund et al., 2006). There is also a growing evidence for a stronger Siberian High prevailing from 1450 CE to 1900 CE based on increased $Na^{2+}$ content in the GISP2 record from Greenland (Mayewski et al., 1997; Meeker and Mayewski, 2002). The onset of the LIA (~1350 CE) on the Swedish west coast also coincided in time with an outbreak of Black Death, which decreased the population by

50-60% and resulted in large-scale farm abandonment with negative implications for land use (Harrison, 2000).

    For the Gullmar Fjord a general cooling during the LIA has been previously suggested based on increased abundances of cryophilic dinocysts (Harland et al, 2013) and benthic foraminifer *Adercotryma glomerata*, which prefers bottom water temperatures < 4°C (Polovodova Asteman et al., 2013). This agrees rather well with the data presented herein, which show temperatures as low as ~3.4 – 4.4°C around 1350 CE, 1500 CE, 1550 CE and 1700-1850 CE with a general temperature

range of 2.9-6.6°C for the whole LIA period (Fig. 5). Based on foraminiferal faunal and $\delta^{13}C$ data Polovodova Asteman et al. (2013) divided the LIA into two distinct phases in the Gullmar Fjord: 1) 1350 –1650 CE and 2) 1650 –1850 CE separated by a short-lived warming centred at ~1650 CE. The reconstructed temperatures show as well a short milder episode based on positive anomalies between ~1570 and 1700 CE (Fig. 6: see pink box). Similar warm, but slightly displaced in time, event is visible in other climate records (Fig. 6) from the North Atlantic and northern hemisphere (Cronin et al., 2003; Moberg et al.,

2005; Cage and Austin, 2010; Hald et al., 2011) suggesting that this short-lived warming was a larger-scale phenomenon possibly linked to a strengthening of the winter NAO, which might have enhanced the AMOC (Cage and Austin, 2010). Indeed, several studies report a long-lasting warm conditions in Europe associated with year 1540 (Casty et al., 2005; Pauling et al., 2006; Wetter et al, 2014), which given our age model uncertainty (±40 yr, see Table 2) for the time interval 1538-1664 CE may well fall within the warm period identified for the LIA from our BWT record. A warming around 1540 is

also seen in winter temperature reconstruction for Stockholm ports and harbours based on historical records of sea ice (Leijonhufvud et al., 2009). The model-based reconstruction by Orth et al (2016) suggests that the European temperatures of 1540 exceeded those of the summer 2003, which was likely the warmest for centuries (e.g. Luterbacher et al, 2016). This is, however, difficult to deduce based on data presented herein, since i) the fjord BWT represent winter temperatures and ii) the record stretches only until ~1996.

The climax or the coldest part of the LIA is often linked to the Maunder minimum in solar activity, which occurred at ~1645–1715 CE (Mauquoy et al., 2002). Our record shows a distinct cooling at around 1750 CE with temperatures ~1°C below the 1961-1999 mean, which given a calibrated $^{14}C$ age range for this particular date (1675-1813 CE ±25 years: see Table 2), may well represent the Maunder minimum in our record. At the same time, a 500-yr long reconstruction of

Stockholm winter temperatures based on sea ice records from local ports and harbours does not show the coldest temperatures during the LIA climax demonstrating instead that the coldest decade for the last 500 years occurred during 1592-1601 CE with average negative temperature anomalies of ~ -4°C (Leijonhufvud et al., 2009).

It appears rather intriguing that the coldest bottom water temperatures for the last 2500 years in the Gullmar Fjord are associated with the onset of the LIA (1350 CE, ~2°C colder than the 1961-1999 mean) rather than with its climax (Figs 5-6). This agrees well with the LIA temperature evolution reported for Loch Sunart (Cage and Austin, 2010) and Chesapeake Bay (Cronin et al., 2003), which both show 2-4°C cooling of the bottom waters at the MCA-LIA transition (Fig. 6), attributed to a switch from the positive winter NAO mode dominating during the medieval times (e.g. Trouet et al., 2009; Faust et al., 2016) to the negative NAO prevailing during the major part of the Little Ice Age. Such a switch in the NAO has been linked to a relaxation of the persistent La-Niña – like conditions in the equatorial Pacific dominating the MCA (Trouet et al., 2009). The MCA-LIA transition has been dated to 1250 CE (Cunningham et al., 2013), 1400 CE (McGregor et al., 2015) and 1450 CE (Cage and Austin, 2010), in contrast to our study (1350 CE), which may again be a result of [14]C dating uncertainties valid for all of the above-mentioned marine records. At the same time the Chesapeake Bay study places MCA-LIA transition in between 1300 and 1400 CE (Cronin et al., 2003), which agrees with our data.

Another interesting feature of the LIA climate variability is associated with consistently low fjord BWT as well as reduced air temperatures during 1790 – 1820 CE as indicated by Stockholm and Central England instrumental time series (Fig. 8). Despite this time period is known to coincide with the Dalton minimum in solar activity (Grove, 1988), it is likely that volcanic activity played much more important role in climate cooling (e.g. Wagner and Zorita, 2005, McGregor et al., 2015). The role of AMOC strength in shaping the LIA cold periods is also somewhat controversial based on marine geological evidence: though the AMOC weakening was proposed as a trigger for the LIA cooling (Bianchi and McCave, 1999), it was argued against (Keigwin and Boyle, 2000) and was not statistically significant in paleoclimate modelling (Van der Schrier and Barkmeijer, 2005). It has even been suggested that Gulf Stream may have experienced warming during this period (e.g. Keigwin and Pickart, 1999), which certainly does not explain low BWT temperatures in our record, as well as low air temperatures over Stockholm and Central England during 1790 – 1820 CE. An explanation for this phenomenon has been proposed by Bjerknes (1965), who postulated, "a decrease in western European winter surface air temperatures during 1790 – 1820 CE to be related almost completely to an anomalous southward advection of cold polar air", a hypothesis later supported by a model study of Van der Schrier and Barkmeijer (2005).

**5.5 The Contemporary Warm Period (~ 1850 CE – 1996)**

Most of the proxy records in the North Atlantic indicate a clear warming trend for the last 100-200 years (Hald et al., 2011 and references therein) similar to our data picking up the warm 1930s and the 1990s (Fig. 8). The 500-yr long reconstruction of Stockholm winter temperatures also demonstrates that the 20[th] century has experienced four out of five warmest decades over the last 500 years: 1905-1914, 1930-1939, 1989-1998 and 1999-2008 (Leijonhufvud et al., 2009). Gullmar Fjord

temperature record shows that when considering a 3-point running mean temperature variability, the most recent warming does not stand out in comparison to the RWP and the MCA, as it has been also previously demonstrated by other studies such as e.g. a tree ring-based summer temperature record from central Scandinavia (Linderholm and Gunnarson, 2005), the Scottish loch data (Cage and Austin, 2010), the North Atlantic SST composite (Cunningham et al., 2013), and a 2000-year

temperature record for continental Europe (PAGES2K, 2013). At the same time, such a "not outstanding recent warming" seen in our dataset is in contrast with the Malangen Fjord record (Hald et al., 2011), according to which the last 100 years are the warmest in the last two millennia. This may reflect the so-called polar amplification, as suggested by the authors, since Malangen Fjord is located much more to the north than Loch Sunart and Gullmar Fjord, both comparably temperate fjord inlets. On the other hand, the stronger recent warming of the Norwegian fjord record may also be explained by a more direct

link to the northward flow of the Atlantic water as compared to Gullmar Fjord, which is i) not located within the core of the North Atlantic Current (Fig.1) and ii) reflects temperature variability during the winter season. At the same time the spring SST reconstruction from the Chesapeake Bay (Cronin et al., 2003, Fig. 6 herein) shows that the 20th century warming clearly exceeds temperatures observed during the prior 2500 years. The shallow Chesapeake Bay displays large seasonal temperature and salinity variability (Cronin et al., 2003) in contrast to Gullmar Fjord, Malangen Fjord and Loch Sunart,

which all have slightly/ less variable bottom water conditions during the year and similar "fjordic" circulation with annual or less frequent basin water exchanges. Also the SST record from the Chesapeake Bay is the shallowest temperature reconstruction (12-25 m w. d.) among the temperature records considered herein (Loch Sunart: 56 m; Gullmar Fjord: 120 m and Malangen Fjord: 218 m w. d.). Shallow water areas are known to generally warm up faster, especially given the facilitating atmospheric warming of the late 20th century due to increase of greenhouse gas emissions (e.g. Masson-Delmotte,

2013), which also may explain why the recent SST increase in the Chesapeake Bay record is unprecedented in a 2500-year perspective.

Studying the instrumental hydrographic time series from the Gullmar Fjord plotted versus reconstructed temperatures (Fig. 7) makes it clear that our record captures the most recent warm period with the bottom water temperatures, which increased by ~1.5°C since the 1960s. Similar increase has been documented for Loch Sunart (Cage and Austin, 2010) and

Ranafjorden, NW coast of Norway (Klitgaard-Kristensen et al., 2004). Instrumental meteorological time series for air temperatures since 1960s from Stockholm and the Central England also demonstrate a winter temperature increase by 3-3.5°C, which is higher than the reconstructed range of Gullmar Fjord bottom water temperatures for this period (Fig. 8). Overall, the variability in reconstructed fjord temperatures corresponds well with both meteorological datasets from 1750 to 1990, by an exception of individual wiggle mismatch between 1930 and 1990 (Fig. 8). In general, it appears that for the

1930-1990 period both air temperatures records lead the observed variability while bottom water temperatures are lagging behind (Fig. 8).

Our record also shows higher BWT prior to the 1920s (Fig. 8), which coincides with the cold AMO (low SSTs) and low sea surface salinities in the North Atlantic and Subpolar Gyre (Reverdin et al. 1994; Reverdin, 2010), while in the following period until ~1960, the reconstructed BWT remains at a lower level (during the warm AMO, i.e. high North Atlantic SSTs),

after which it peaks again at time of "Great Salinity Anomaly" during the late 1970s and late 1980s (Dickson et al., 1988; Belkin et al., 1998). It remains intriguing, though, that at both occasions (prior to the 1920s and during the 1970s/1980s) of the reduced salinities and low SSTs in the North Atlantic, our record is characterized by high temperatures of the fjord deep water, which is consistent with increasing air temperatures in instrumental datasets from Stockholm and Central England

(Fig. 8). The low surface salinities of the Great Salinity Anomaly were likely driven by an increased freshwater/sea ice export from the Arctic via Fram Strait and Canadian Archipelago (Belkin et al., 1998). The increased freshwater flux into the subpolar North Atlantic, in turn, is suggested to increase salinity of the North Atlantic Current, which may reduce its predicted weakening due to enhanced freshwater fluxes and will help to restart a stronger AMOC (Hátún et al., 2005; Thornalley et al., 2009). A stronger North Atlantic Current would in turn result in an increased heat transport during winter

to the Eastern North Atlantic and together with other external forcing factors (e.g. changes in NAO, volcanism, and solar activity) would contribute to the warming observed in the fjord BWT record during the early 20[th] century. One of those factors, the positive NAO mode, which prevailed since the 1970s/1980s (Hurrell, 1995; http://www.cpc.ncep.noaa.gov/products/precip/CWlink/pna/season.JFM.nao.gif), extracts heat from the subpolar North Atlantic through increased westerlies over that region, decreases SSTs, enhances convection, increases ocean density

(Delworth et al., 2016; Delworth and Zeng, 2016) and results in milder winter conditions over the north-western Europe, thus counteracting effects of the AMOC weakening, which has been suggested for the 20[th] century based on modeling data and proxy records (Caesar et al., 2018; Thornalley et al., 2018). Also, located within a coastal region, the Gullmar Fjord is more susceptible to wind-forced temperature changes, which follow the variability of the NAO index and drive coastal upwelling and downwelling in the fjord (Björk and Nordberg, 2003). According to Jansen et al. (2007), the late 20[th] century

warming as demonstrated by many proxy records from the NE Atlantic (see discussion above), is unlikely to be explained by the external forcing factors and is probably linked to the anthropogenic drivers such as greenhouse gas emissions and aerosols (Booth et al, 2012), which both significantly increased since ~1970s (Masson-Delmotte, 2013).

When studying the Gullmar Fjord bottom water temperature record for the last 2500 years, it is interesting to note that the most recent warming of the 20[th] century (presented herein until 1996) does not stand out but appears to be comparable to

both the Roman Warm Period and the Medieval Climate Anomaly. This observation has, however, to be used with caution since our dataset does not go beyond year 1996 due to a lack of material (see discussion below) and, hence, does not cover the most recent part of the 20th century warming, widely accepted as triggered by growing anthropogenic emissions.

**5.6 Environmental conditions explaining absence or rare occurrence of *Cassidulina laevigata* in the record.**

Since 1990, *Cassidulina laevigata* has dramatically decreased in abundances in the Gullmar Fjord deep basin (Fig. 6). A

similar pattern, with short disappearances of *C. laevigata*, is seen during the Roman and Medieval warm periods (Fig. 6). This effect may be either due to the increased temperatures and/or, more likely, due to periods of severe hypoxia as *C. laevigata* is documented to be sensitive to oxygen concentrations below 1 ml l$^{-1}$ (e.g. Gustafsson & Nordberg 2001; Nardelli

et al., 2014). To a large extent, the oxygen status of fjords and estuaries on the Swedish west coast, is controlled by climate (e.g. Nordberg et al., 2000; Filipsson and Nordberg 2004a, b), but the late Holocene changes in land use and organic enrichment in the fjord are also suggested to play a role (Filipsson and Nordberg, 2010). Thus, the short extinctions of *C. laevigata* during warmer periods further back in time may be equivalent to the present-day pattern of severe hypoxia following the positive NAO periods with mild and humid winters, limited basin water exchange and high organic matter flux increasing oxygen demand (Nordberg et al. 2000, 2001; Filipsson & Nordberg 2004a). Indeed, when comparing our record to the reconstructed NAO index from the Trondheim Fjord, W Norway (Faust et al., 2016) it appears that sediment core intervals with absent *C. laevigata* (at ~75 CE, 450 CE, 1000 CE and post-1990) correlate rather well with the positive NAO index (Fig. 6).

## 6 Conclusions

To conclude, from the available paleotemperature equations, the equation by McCorkle et al (1997) produced most realistic reconstructed deep water temperature range of 2.7 - 7.8°C, which falls within the annual variability instrumentally recorded in the deep fjord basin since 1890. This suggests that the Gullmar Fjord $\delta^{18}O$ record mainly reflects variability of the winter bottom water temperatures with a minor salinity influence. The relationship between the evolution of the fjord's bottom water temperatures over the last two millennia and other late Holocene climate records reveals synchronous North Atlantic-wide centennial and multidecadal climate variability despite age model uncertainties, different proxy type, time resolution, annual versus seasonal signal and different hydrographic characteristics.

The record shows a substantial and long-term warming during the Roman Warm Period (~350 BCE – 450 CE), followed by variable bottom water temperatures during the Dark Ages (~450 – 850 CE). The Viking Age/Medieval Climate Anomaly (~850 – 1350 CE) is also indicated by positive bottom water temperature anomalies, while the Little Ice Age (~1350 – 1850 CE) is characterized by a long-term cooling with distinct multidecadal variability. The record also picks up the contemporary warming of 1930s and the 1990s. When studying the Gullmar Fjord bottom water temperature record for the last 2500 years, it is interesting to note that the warming of the 20th century (presented herein until 1996) is comparable to both the Roman Warm Period and the Medieval Climate Anomaly.

## Data availability

The data presented in this paper are available at www.pangaea.de.

**Author contribution**

KN conceived the research, obtained funding, as well as, organized and performed sediment core sampling in 1990 and 1999. HLF participated in the 1999 cruise; picked most of the foraminiferal samples, prepared them for stable oxygen isotopes, and funded isotope analysis. IPA participated in additional sampling campaign in 2009, did sediment core sampling, picked and prepared the samples for stable isotope analysis. All authors (IPA, KN and HLF) equally contributed to data analysis & interpretation. IPA wrote the manuscript with the help of both co-authors.

**Competing interests**

The authors declare that they have no conflict of interests.

**Acknowledgements**

The authors sincerely thank everyone who helped to perform this study. The crews of *R/V Svanic*, *R/V Arne Tiselius* and *R/V Skagerak* assisted with sampling. The study was financed by the Swedish Research Council: VR grants no 621-2004-5320, 621-2007-4369 (KN) and 621-2005-4265 (HLF); Lamm Foundation (KN); University of Gothenburg (KN); Marine Research Centre, GMF (KN), and EUROPROX - European Graduate College – Proxies in Earth History (HLF). Monika Segl (University of Bremen) measured stable O and C isotopes. The PALEOSTUDIES program (University of Bremen) covered the costs for isotope analyses, while the Department of Earth Sciences (University of Gothenburg) provided a postdoctoral fellowship to IPA. The manuscript greatly benefited from the insightful comments and suggestions by Antoon Kuijpers, an anonymous reviewer and journal editor Alessio Rovere.

The hydrographic data used in the study were obtained from the SMHI oceanographic observation database (SHARK). The SHARK data collection is organized by the environmental monitoring program and is funded by the Swedish Agency for Marine and Water Management (SwAM).

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

**Figure captions**

**Figure 1:** Map of the study area including location of Gullmar Fjord (GF) and sampling site of Ga113-2Aa & 9004 record (star) within North Atlantic (A) and North Sea – Skagerrak region (B). Locations of other discussed proxy records are shown by white circles, while some of major ocean circulation characteristics mentioned in the text are indicated as: EGC – East

Greenland Current, NAC – North Atlantic Current, SPG – Subpolar Gyre, and STG – Subtropical Gyre (A). B: the major regional water masses and currents are shown as follows: AW - Atlantic Water, SJC – South Jutland Current, NJC – North Jutland Current, BC – Baltic Current, NCC – Norwegian Coastal Current. C: an overview of water column stratification in the longitudinal profile of the Gullmar Fjord with indication of salinity (S) and residence times (t) typical for each water layer (Arneborg, 2004).

**Figure 2:** Hydrographic measurements from Alsbäck Deep, Gullmar Fjord taken during 1890 – 2000 below 110 m water depth: BWT – bottom water temperature (a), salinity (b) and dissolved oxygen (c). A snapshot of hydrographic changes in BWT (d), salinity (e) and oxygen (f) associated with basin water exchanges between 1992 and 1993 showing annual variability of these parameters.

**Figure 3:** Age model of the studied Ga113-2Aa & 9004 record (A) and comparison of foraminiferal and isotopic data with core G113-091, taken at the same location in 2009, to prove the absence of a gap between GA113-2Aa and 9004 (B), according to Polovodova Asteman et al (2013).

**Figure 4:** Comparison of reconstructed temperatures and $\delta^{18}O$ values measured in stained *C. laevigata* from the core tops collected in Gullmar Fjord (G113-091) and the Skagerrak (OS4, OS6, OS14, 9202, 9205) to hydrographic temperature data (A) and to $\delta^{18}O$ predicted from palaeotemperature equation (B) by McCorkle et al (1997). C: Temperature vs. $\delta^{18}Oc - \delta^{18}Ow$, together with the paleotemperature equations from Shackleton (1974), Hays and Grossman (1991), Kim and O'Neil (1997), McCorkle et al. (1997), and Bemis et al. (1998).

**Figure 5:** A 2500-year long $\delta^{18}O$ record (A) and reconstructed winter bottom water temperatures, BWT (B) from Gullmar Fjord. Thick lines show 3-point running mean for both curves, and dashed lines indicate A) a long-term average of 2.4‰ for $\delta^{18}O$ record and B) 5.4°C - a mean for instrumental bottom water temperatures registered between 1961 and 1990. Grey shaded areas in BWT indicate a median offset (0.7°C) in instrumental versus reconstructed temperatures obtained for rose Bengal stained *C. laevigata* from the core tops (see Fig. 4A), used herein as an error margin. C: Box and whisker plot shows a range for instrumental BWT observations performed during 1890 – 1999 and measured at more regular intervals from the 1960s, the data is from water depths ≥110 m in the fjord deepest basin (Alsbäck Deep). The middle, the upper and the lower horizontal lines in the box indicate the median, 75 and 25 percentiles, respectively. Abbreviations are as follows: RWP – the Roman Warm Period, DA – the Dark Ages, VA/MCA – the Viking Age/Medieval Climate Anomaly and LIA – the Little Ice Age.

**Figure 6:** Reconstructed bottom water temperatures (BWT) shown as anomaly against the 1961-1990 instrumental mean of

5.4°C from Gullmar Fjord compared against other temperature proxy records: annual northern hemisphere temperatures (Moberg et al., 2005), bottom water temperatures from Malangen Fjord in NW Norway (Hald et al., 2011) and Loch Sunart in Scotland (Cage and Austin, 2010), spring sea surface temperatures from Chesapeake Bay, E North Atlantic Ocean (Cronin et al., 2003), annual temperatures reconstructed for continental Europe (Pages2K, 2013) and the reconstructed NAO record from Trondheim Fjord, W Norway (Faust et al., 2016). Also are shown relative abundances of foraminifer *Cassidulina laevigata* in the fjord with abundance minima and respective gaps in temperature reconstruction linked to the positive NAO index (arrows). For location of these proxy records see Fig. 1A and for abbreviations see text to Fig. 5. Grey shaded areas in Gullmar Fjord BWT anomalies indicate a median offset (0.7°C) in instrumental versus reconstructed temperatures (see Fig. 4A) obtained for rose Bengal stained *C. laevigata* from the core tops, used herein as an error margin. Blue and pink boxes depict discussed in the text a short-lived cooling at ~1250 CE and a warm interval between ~1570 and 1700 CE, respectively.

**Figure 7:** Comparison of the winter bottom water temperatures (BWT) reconstructed from Gullmar Fjord record to instrumental basin water temperatures measured in the deepest fjord basin: the annual mean (a), mean for May-August (b) and mean for January-March (c).

**Figure 8:** Comparison of reconstructed winter bottom water temperatures (BWT) from Gullmar Fjord to meteorological observations of winter air temperatures recorded for Stockholm (stippled line) and the Central England (solid line without symbols).

**Supplementary Figure 1:** Scatter plot of stable carbon isotopes ($\delta^{13}$C) data from the composite G113-2Aa – 9004 record (Filipsson and Nordberg, 2010) plotted against the oxygen isotope data presented herein. Note absence of correlation between the two, ruling out the possibility that the changes in $\delta^{18}$O are due to changes in water masses

**Table captions:**

**Table 1:** Stations with collected sediment core tops and $\delta^{18}$O analyzed on living (rose Bengal stained) *Cassidulina laevigata*.

**Table 2.** AMS $^{14}$C dates obtained for the gravity core 9004 and calibrated calendar ages. All dates presented in Filipsson and Nordberg (2010) and Polovodova Asteman et al. (2013) were re-calibrated using Calib 7.10 (Stuiver *et al.* 2017), the Marine13 calibration dataset (Reimer et al, 2013), and ΔR = 100 ± 50. Asterisks (*) show dates not used in the final age model due to age reversals.

Figure 1

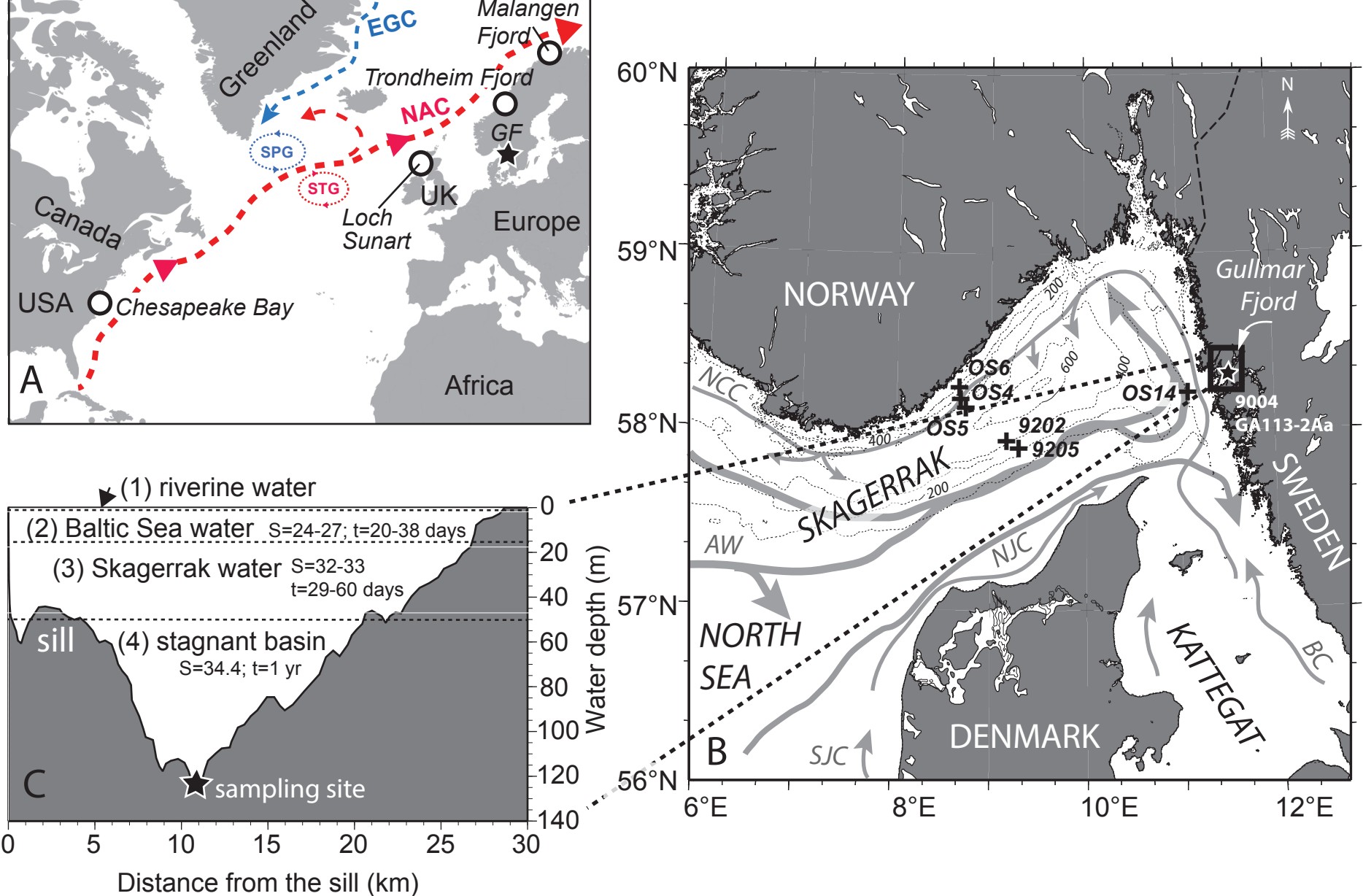

# Figure 2

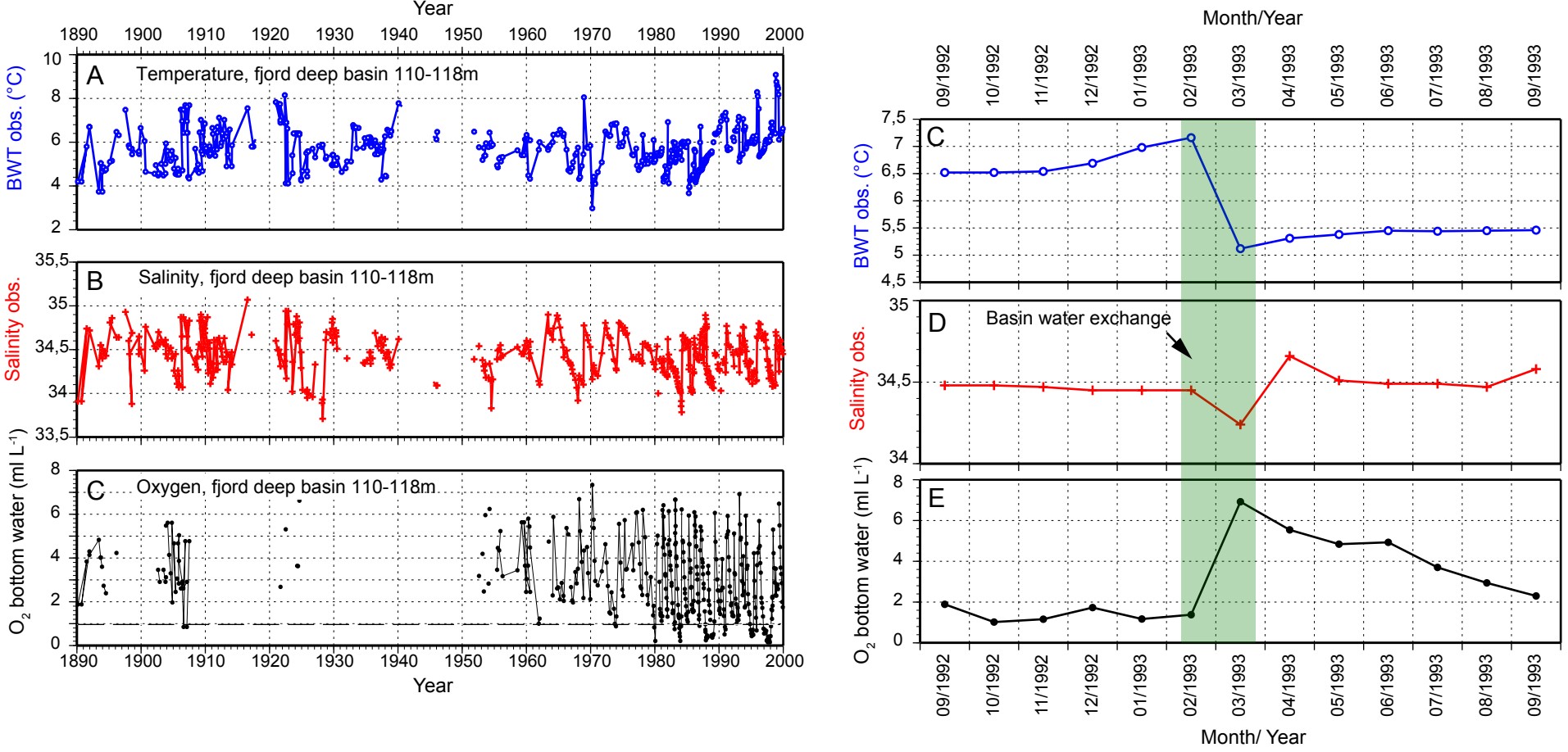

Figure 3

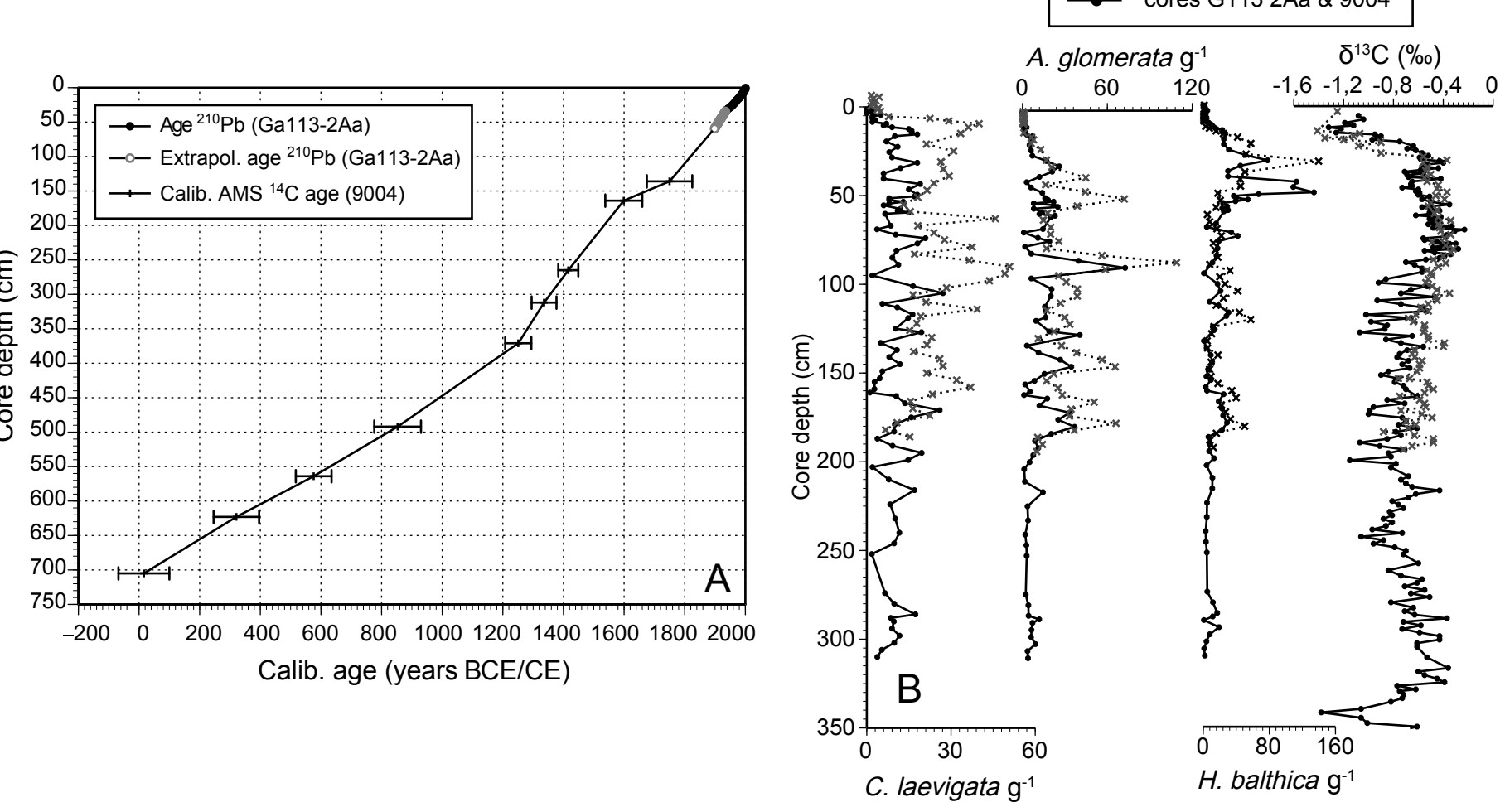

# Figure 4

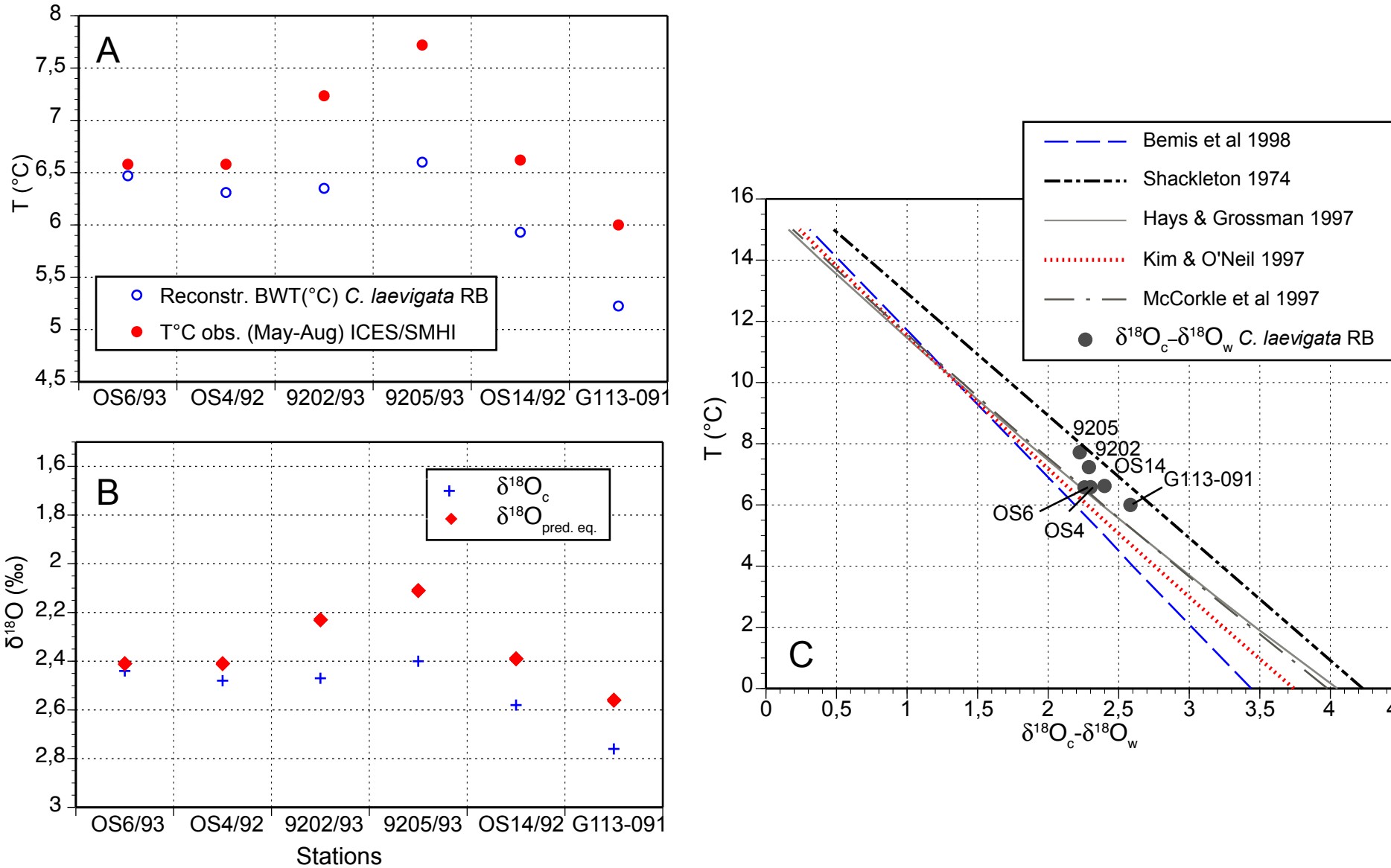

Figure 5

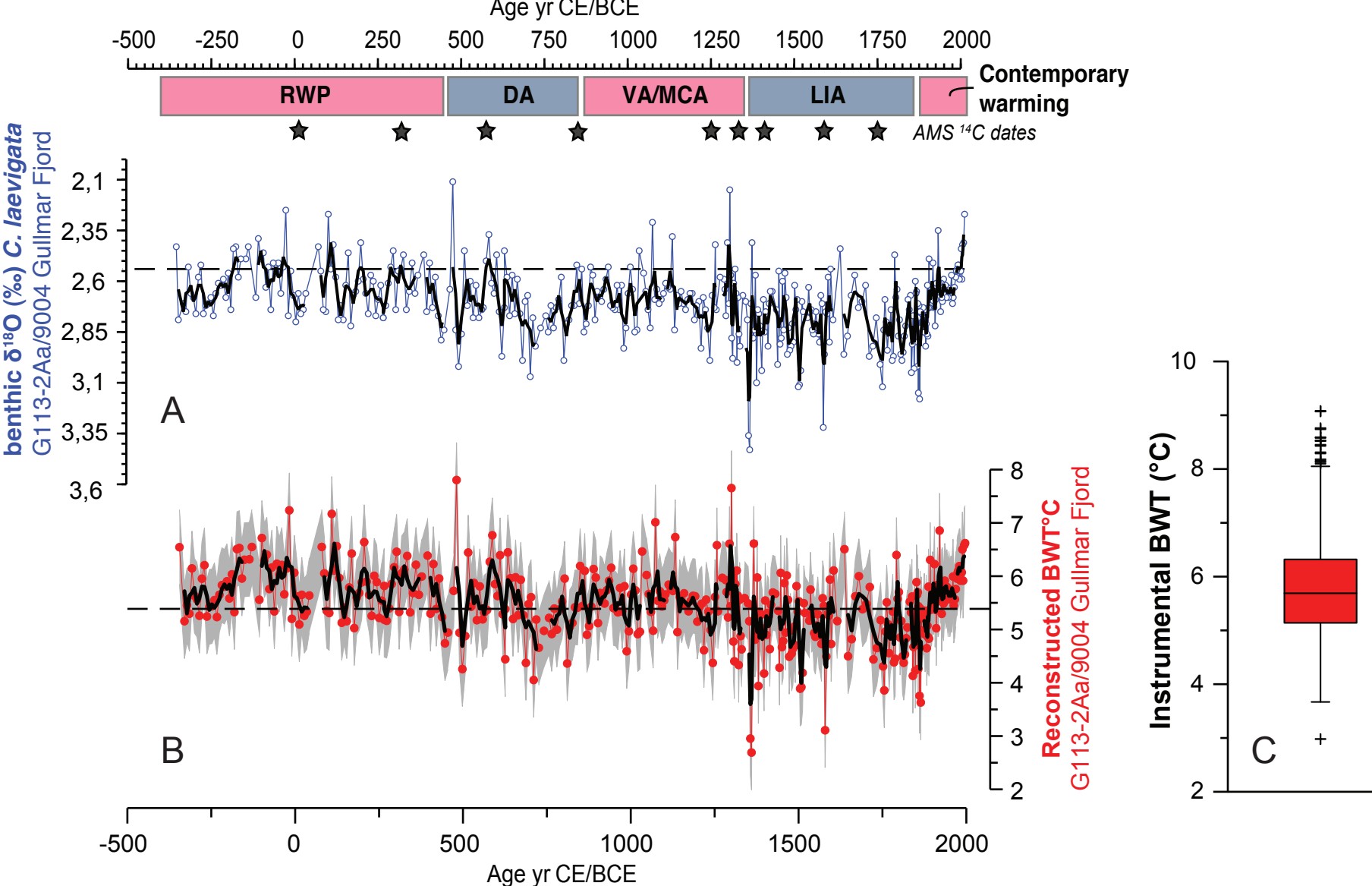

Figure 6

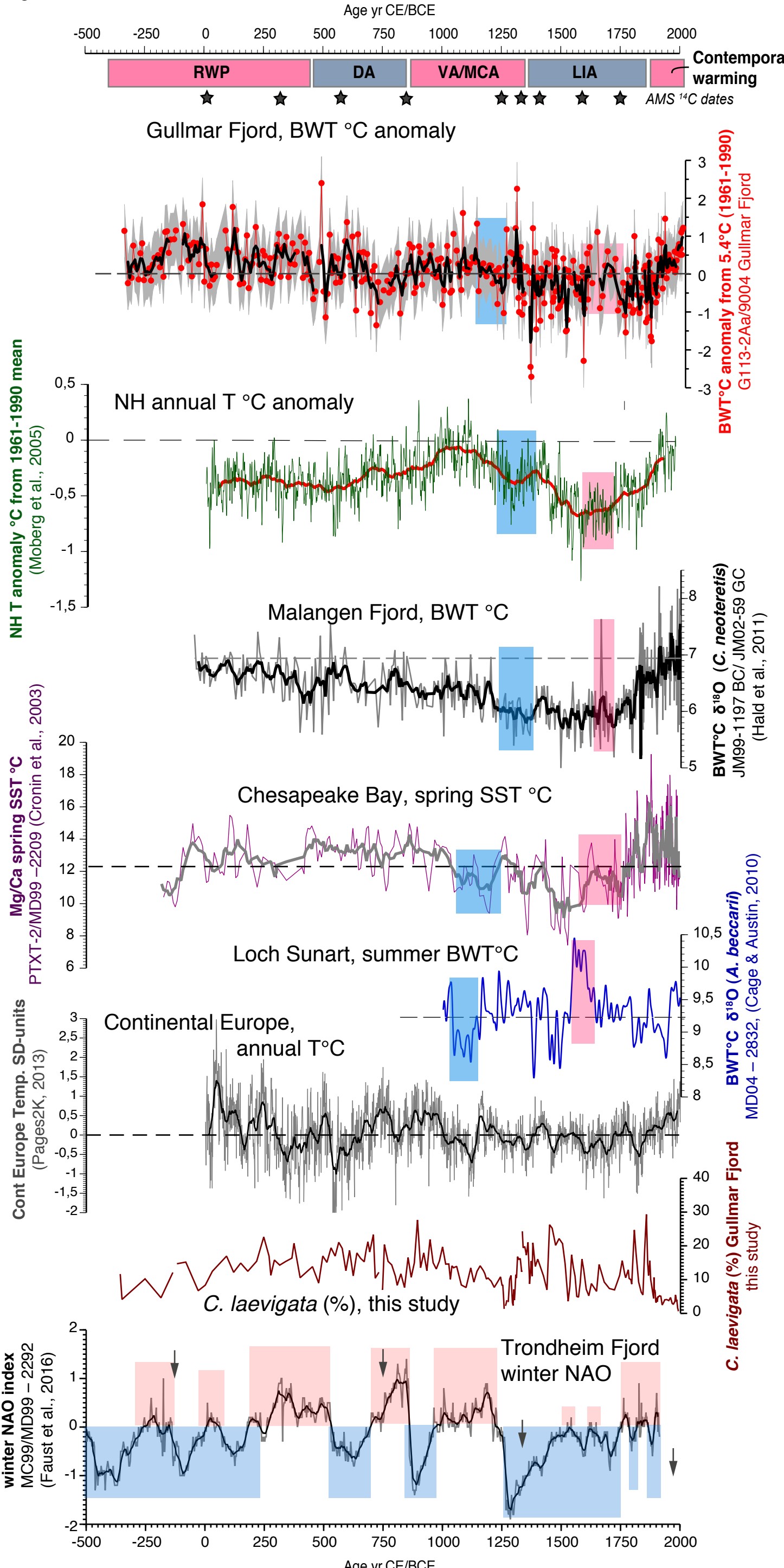

Figure 7

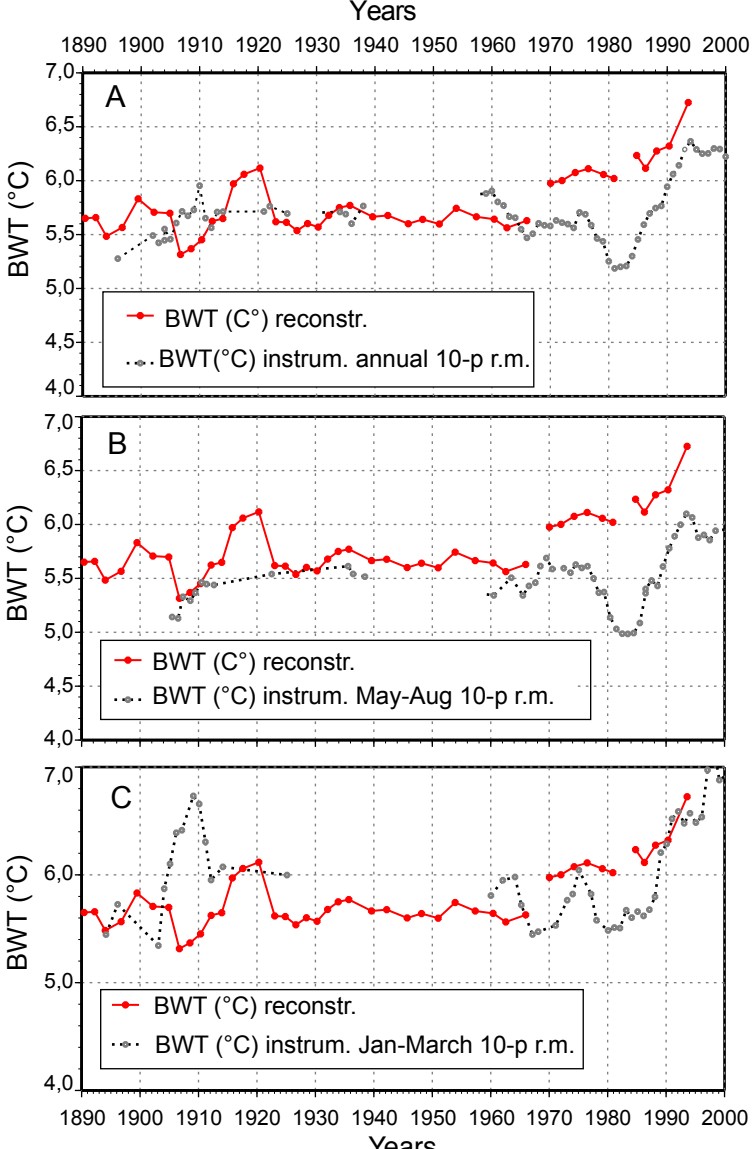

Figure 8

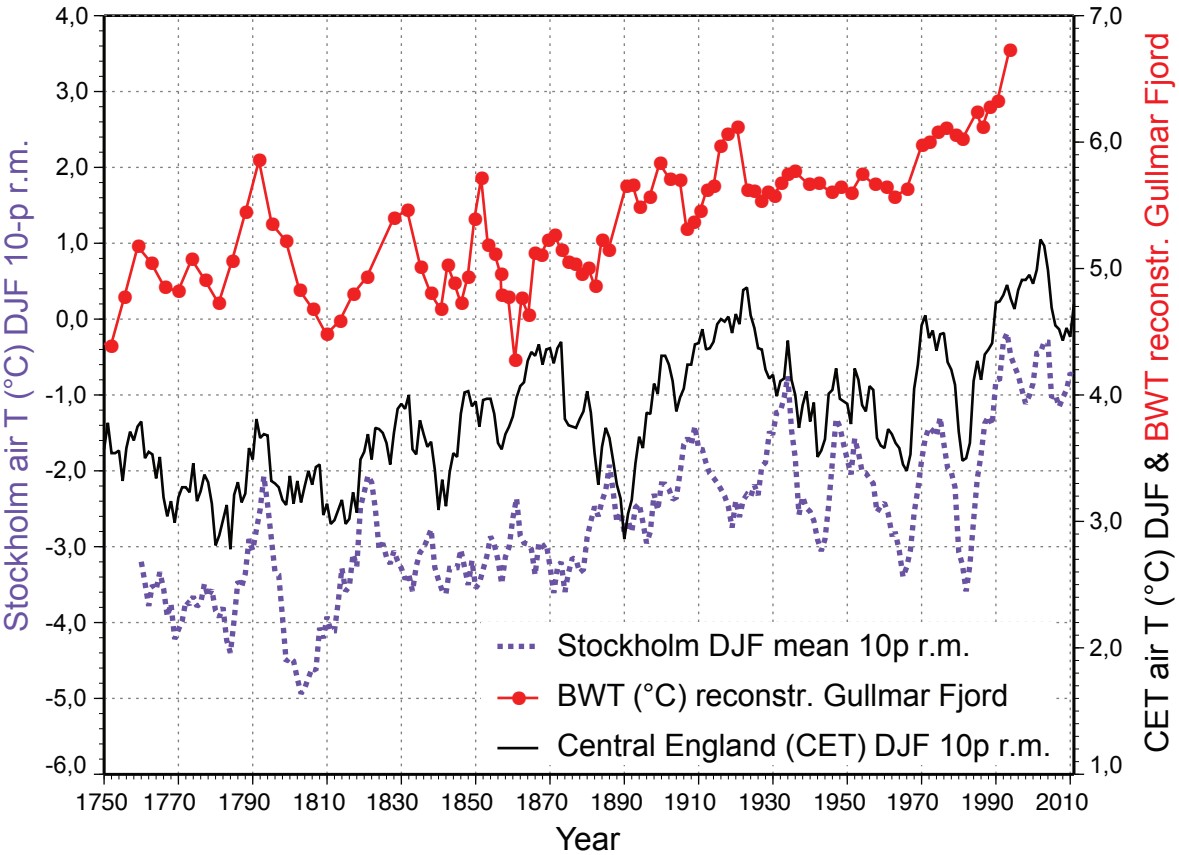

Supplementary Fig. 1

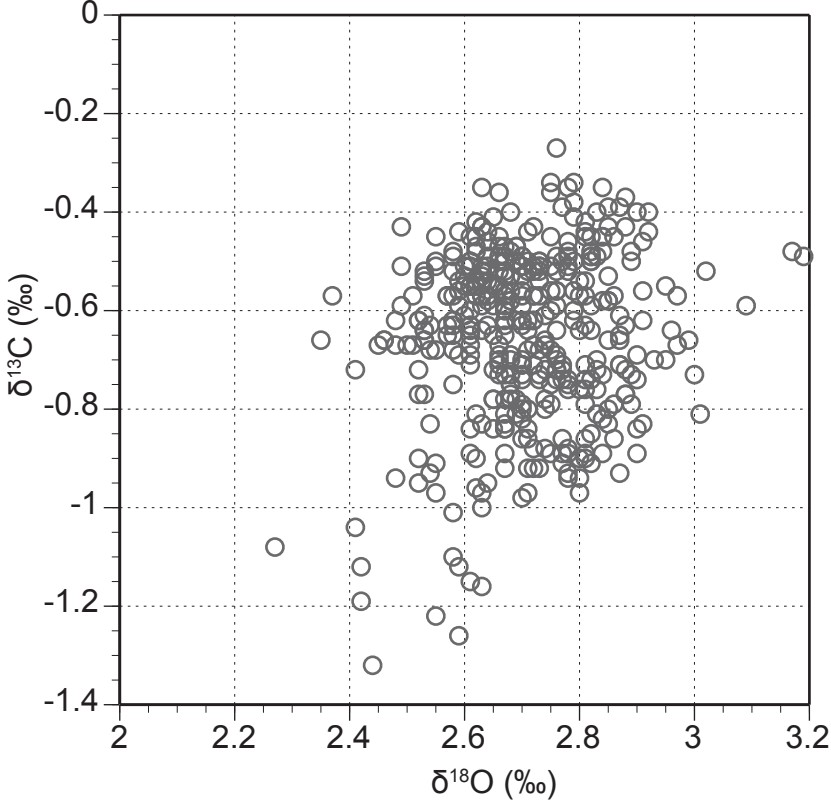

**Table 1**: Stations with collected sediment core tops and $\delta^{18}$O analyzed on living (rose Bengal stained) *Cassidulina laevigata*.

| Station | Latitude N | Longitude E | Water depth, m | Sampling date | $\delta^{18}$O, ‰ |
|---------|-----------|-------------|----------------|---------------|-------------------|
| 9202 | 57°56.2' | 9°27.3' | 177 | 1992-08-04 | 2.49 |
| 9202 | 57°56.2' | 9°27.3' | 177 | 1992-08-04 | 2.44 |
| 9205 | 57°58.4' | 9°24.0' | 294 | 1992-08-06 | 2.40 |
| OS14 | 58°06.06' | 10°58.27' | 135 | 1993-05-09 | 2.58 |
| OS4 | 58°18.54' | 8°54.99' | 325 | 1993-05-04 | 2.48 |
| OS6 | 58°21.58' | 8°51.01' | 177 | 1992-08-04 | 2.43 |
| G113-091 | 58°17.570' | 11°23.060' | 116 | 2009-09-01 | 2.76 |

**Table 2.** AMS ¹⁴C dates obtained for the gravity core 9004 and calibrated calendar ages. All dates presented in Filipsson and Nordberg (2010) and Polovodova Asteman et al. (2013) were re-calibrated using Calib 7.10 (Stuiver *et al.* 2017), the Marine13 calibration dataset (Reimer et al, 2013), and ΔR = 100 ± 50. Asterisks (*) show dates not used in the final age model due to age reversals.

| Core | Core depth (cm) | Lab. ID | Dated bivalve species | ¹⁴C age (years BP) | Error (±) | Calibrated age range, ±1σ, ΔR=100±50 (years CE/BCE) | Relative probability | Calibrated age, median probability (years CE) |
|---|---|---|---|---|---|---|---|---|
| 9004 | 98 | Ua-24043 | *Nuculana minuta* | 710* | 35* | 1645-1806* | 1 | 1702* |
| 9004 | 136 | Ua-35966 | *Nuculana pernula* | 675 | 25 | 1675-1813 | 1 | 1750 |
| 9004 | 164 | Ua-23075 | *Yoldiella lenticula* | 800 | 40 | 1538-1664 | 1 | 1599 |
| 9004 | 265 | Ua-35967 | *Nucula* sp. | 1025 | 30 | 1356-1372/1383-1465 | 0.106/0.894 | 1416 |
| 9004 | 312 | Ua-35968 | *Clamys septemradiatus* | 1145 | 25 | 1295-1389 | 1 | 1336 |
| 9004 | 313 | Ua-23000 | *Abra nitida* | 1305* | 45* | 1138-1276* | 1 | 1195* |
| 9004 | 371 | Ua-35969 | *Nucula tenuis* | 1245 | 25 | 1208-1303 | 1 | 1251 |
| 9004 | 492 | Ua-23001 | *Abra nitida* | 1640 | 45 | 776-938 | 1 | 853 |
| 9004 | 564 | Ua-23002 | *Nuculana minuta* | 1925 | 40 | 517-658 | 1 | 576 |
| 9004 | 623 | Ua-23003 | *Thyasira flexuosa* | 2155 | 45 | 246-410 | 1 | 321 |
| 9004 | 705 | Ua-23004 | *Thyasira flexuosa* | 2415 | 45 | 68 BCE-102 CE | 1 | 16 |