# Peer review of "Tracing winter temperatures over the last two millennia using a NE Atlantic coastal record"

_Climate of the Past, 2017_

## Referee Comment (RC1) · A. Kuijpers (Referee) · 12 Jan 2018

Excellent contribution, well-written and high-standard figures.

A few minor comments requiring a few more, additional lines of discussion: *) comment/discuss general results in context of (non-cited) highly relevant reference Luterbach et al. 2016 Env.Res.Lett. 11.

*) start of LIA : refer to Stuiver et al. 1995, Quat Res 44

*) multi-decadal variability lacking reference to possible link to Atlantic Multidecadal Oscillation (AMO). Within this context interesting to discuss results shown in Fig. 7

with peaking BWT values prior to 1920 coinciding with cold AMO / low N Atlantic sea surface salinities (Reverdin et al. 1994 Progr.Ocean. 33; Reverdin 2010, Journ Clim 23) , in following period until ca 1960 BWT at a lower level (during warm AMO) , after which again peaking (e.g. at time of 'Great Salinity Anomaly', early 1970's).

*) Fig. 8 Discuss Dalton Minimum (AD 1790 - 1820) with general low T, both Tair and BWT coinciding with Gulf Stream warming ( see previous remark !) , ref Van der Schrier and Barkmeijer 2005, Clim Dyn. 24

---

## Referee Comment (RC2) · Anonymous Referee #2 · 15 Feb 2018

**Irina Polovodova Asteman, Helena L. Filipsson and Kjell Nordberg: Tracing winter temperatures over the last two millennia using a NE Atlantic coastal record. Climate of the Past Discussions.**

The manuscript presents a record of environmental and climate change from the Eastern North Atlantic region. The data are of good quality and although some local conditions may be expected, the authors argue well that the record does indeed relate to more regional conditions. One of the most interesting aspects is that the record may indeed represent winter conditions. The subject fits well to Climate of the Past and overall the manuscript is well written, although the general significance of the data could be made clearer and some additional explanations are in order. I thus recommend publication following moderate revisions.

**1. ) Relevance:** The authors need to explain better why the study is important. It is mentioned in the introduction that only few high-resolution records of late Holocene conditions exist from the eastern North Atlantic region. But records also exist from other regions, both Iceland and the western North Atlantic and the Labrador Sea region. Why is the Eastern North Atlantic region important? Please add a short explanation, what is special/different about this region compared to other areas. How can this study improve our general understanding of the late Holocene climate of the North Atlantic and which mechanisms control climate and ocean variability?

**2) Bottom Water Tempertures:** Page 3, line 28-31. It is stated that the water exchange only occurs during winter. Does any change in salinity or temperature conditions of the bottom waters occur during spring/summer?

Explain more clearly whether the Bottom Water Temperatures actually represent winter conditions (mention this also in the abstract). As this is a central part of the work, it needs to be explained very clearly.

Seasons used in the bottom-water temperature reconstruction (p9): Traditionally the winter season is described through the months DJF, but here the period JFM is used. Why? Is there a local environmental reason for this, purely due to available data, or …? Similarly an explanation should be given for the use of May-August as the summer period, but this is normally JJA. It is not directly stated in paragraph 4.3 that these periods correspond to "winter" (JFM) and "summer" (MJJA), but in the following discussion (paragraph 5) winter temperatures are mentioned, so I assume that this is the case? However, it needs to be stated clearly and explained properly

**3) General interpretation and potential link to the North Atlantic Oscillation: 3a)** The section on the influence of the North Atlantic Oscillation (NAO) on the Gullmar Fjord (p 11, lines 1.6) should be moved to the introduction, with reference also to modern data from NE Europe/NE Atlantic. No reference to NAO during past climate periods should be mentioned as fact before this is discussed in the following paragraphs.

**3b)** The potential role of the NAO is discussed for the MCA and LIA. But what about the RWP and the DACP? Several studies have indicated that climate during these periods may also be linked to the NAO, and the manuscript would benefit form a more in-depth discussion – and reference to a wider range of previously published studies. It is also noteworthy that the authors only refer to work that shows comparable conditions as seen in Gullmar Fjorden, omitting any other studies. The authors should also look towards studies on the Late Holocene from further afield, e.g. Portugal, East Greenland, the Labrador Sea.

**4) Hypoxia:** On P. 7, line 5 and again Page 10, line 17-18 it is mentioned that *C. laevigata* has become a rare species in the Gullmar Fjord since 1990. One page 7 no explanation is given, page 10 the phenomenon is explained through hypoxia. However on page 15 a discussion is raised, whether it is due to hyposix and if yes, why. The discussion is certainly relevant but the fact that first a statement is made and later a discussion is raised, makes it confusing and somewhat messy.

I would suggest just to refer to "see discussion" instead of jumping the gun on p.7 and 10. Also, the discussion on p 15 does not really fit well to the remaining text, and a solution may be to move this hypoxia discussion to its own, separate paragraph.

With respect to this discussion, the authors basically explain the hypoxia as due to climate change. However, what about the increased nutrient supply seen due to more intensified farming seen in the general region, may this also play a role? Please discuss.

**5) Conclusion paragraph:** The paragraph should be expanded with a synopsis on the discussion on the processes driving the climate change.

**Minor comments:**

Foraminiferal species: add author name to the species name the first time a species is mentioned: i.e., *Cassidulina laevigata* d'Orbigny, 1826; *Adercotryma glomerata* (Brady, 1878); *Hyalinea balthica* (Schröter in Gmelin, 1791).

P5, line 14; reservoir correction: How many bivalve shells and from many sites in the Gullmar Fjord is is reservoir correction based on?

P.9, line 24: add reference for timing of the foraminiferal growth season.

P10, line 22-25: add references for the mentioned climatic intervals.

Page 15: Could the stronger recent warming of the Marlangen Fjord region be due to a more direct link to the northward flow of Atlantic water compared to Gullmar Fjord, which is not in direct contact with the core of the Atlantic water?

**Figures and figure captions**:

All terms and abbreviations should be explained.

Fig 1: explain abreviations for current names. Fig 1a: land masses are shown in a very pale gray – it would be easier to see, if landmasses were shown in a slightly darker colour.

Fig. 2: BWT needs to be explained wither in the figures or the figure captions, as it should be possible to understand the figures without reading the main text.

Fig 3A: I cannot distinguish between the upper and lower symbol; please make them more different.

Fig. 5: explain BWT, RWP, DA, LIA etc in the figure caption. Mark the present BWT range on the figure.

Fig. 6: explain the pink and blue inervals.

Fig. 7: Here "bottom water temperature" is written in full (not giving the abbreviation) – be consistent.

Some additional comments are provided as comments in pdf file of the manuscript (only relevant pages).

[revised manuscript text omitted]

---

## Author Comment (AC1) · 18 Apr 2018

Author's response to reviewer 1

We sincerely thank reviewer 1, Antoon Kuijpers, for his useful and insightful comments and an advice on highly relevant references, which we missed to include. Below we respond to the comments point by point:

Comment 1: Comment/ discuss general results in context of (non-cited) highly relevant reference Luterbacher et al. 2016 Env.Res.Lett. 11.

Response: We first had troubles tracking down the suggested reference, until we have

found out that it must be a paper by Orth et al (2016), including Luterbacher as a co-author, since it appears to be the only paper in Env.Res.Lett. 11, which, indeed, turned out to be highly relevant for our discussion. Now we included this into the discussion and added the following section under the "LIA milder episode": ÂńIndeed, several studies report an exceptional multi-month drought and long-lasting warm conditions in Europe associated with year 1540 (Casty et al., 2005; Pauling et al., 2006; Wetter et al, 2014), which given our age model uncertainty for the time interval 1538-1664 BCE (+-40 yr, see Table 2) may well fall within the warm period identified for the LIA from our BWT record. A warming around 1540 is also seen in winter temperature reconstruction for Stockholm ports and harbours based on historical records of sea ice (Leijonhufvud et al., 2009). The model-based reconstruction by Orth et al (2016) suggests that the European temperatures of 1540 exceeded those of the summer 2003, which was likely the warmest for centuries (e.g. Luterbacher et al, 2016). This is, however, difficult to deduce based on our data, since the fjord BWT record only stretches until ca 1996.Âż

Comment 2: Start of LIA : refer to Stuiver et al. 1995, Quat Res 44

Response: The reference has been included.

Comment 3: Multi-decadal variability lacking reference to possible link to Atlantic Mul-tidecadal Oscillation (AMO). Within this context interesting to discuss results shown in Fig. 7 with peaking BWT values prior to 1920 coinciding with cold AMO / low N Atlantic sea surface salinities (Reverdin et al. 1994 Progr.Ocean. 33; Reverdin 2010, Journ Clim 23) , in following period until ca 1960 BWT at a lower level (during warm AMO), after which again peaking (e.g. at time of 'Great Salinity Anomaly', early 1970's).

Response: We added a reference to AMO (Enfield et al, 2001) and its link to the multidecadal climate variability (through AMOC) into the introduction. We also added a discussion around high fjord BWT at times of cold AMO and reduced salinities in the North Atlantic: "Our record also shows higher BWT prior to the 1920s (Fig. 8), which coincides with the cold AMO (low SSTs) and low sea surface salinities in the North

Atlantic and Subpolar Gyre (Reverdin et al. 1994; Reverdin, 2010), while in the following period until ca 1960, the reconstructed BWT remains at a lower level (during the warm AMO, i.e. high North Atlantic SSTs), after which it peaks again at time of "Great Salinity Anomaly" during the late 1970s and late 1980s (Dickson et al., 1988; Belkin et al., 1998). It remains intriguing, though, that at both occasions (prior to the 1920s and during the 1970s/1980s) of the reduced salinities and low SSTs in the North Atlantic, our record is characterized by high temperatures of the fjord deep water, which is consistent with increasing air temperatures in instrumental datasets from Stockholm and Central England (Fig. 8). The low surface salinities of the Great Salinity Anomaly were likely driven by an increased freshwater/sea ice export from the Arctic via Fram Strait and Canadian Archipelago (Belkin et al., 1998). The increased freshwater flux into the subpolar North Atlantic, in turn, is suggested to increase salinity of the North Atlantic Current, which may reduce its predicted weakening due to enhanced freshwater fluxes and will help to restart a stronger AMOC (Hátún et al., 2005; Thornalley et al., 2009). A stronger North Atlantic Current would in turn result in an increased heat transport during winter to the Eastern North Atlantic and together with other external forcing factors (e.g. changes in NAO, volcanism, and solar activity) would contribute to the warming observed in the fjord BWT record during the early 20th century. One of those factors, the positive NAO mode, which prevailed since the 1970s/1980s (Hurrell, 1995; http://www.cpc.ncep.noaa.gov/products/precip/CWlink/pna/season.JFM.nao.gif), extracts heat from the subpolar North Atlantic through increased westerlies over that region, decreases SSTs, enhances convection, increases ocean density (Delworth et al., 2016; Delworth and Zeng, 2016) and results in milder winter conditions over the north-western Europe, thus counteracting effects of the AMOC weakening, which has been suggested for the 20th century based on modeling data and proxy records (Caesar et al., 2018; Thornalley et al., 2018). Also, located within a coastal region, the Gullmar Fjord is more susceptible to wind-forced temperature changes, which follow the variability of the NAO index and drive coastal upwelling and downwelling in the fjord (Björk and Nordberg, 2003). According to Jansen et al. (2007), the late

20th century warming as demonstrated by many proxy records from the NE Atlantic (see discussion above), is unlikely to be explained by the external forcing factors and is probably linked to the anthropogenic drivers such as greenhouse gas emissions and aerosols (Booth et al, 2012), which both significantly increased since ca 1970s (Masson-Delmotte, 2013)."

Comment 4: Fig. 8: Discuss Dalton Minimum (AD 1790 - 1820) with general low T, both Tair and BWT coinciding with Gulf Stream warming (see previous remark !) , ref Van der Schrier and Barkmeijer 2005, Clim Dyn. 24

Response: We agree with the reviewer and have added a following section into the LIA-discussion:

"Another interesting feature of the LIA climate variability is associated with consistently low fjord BWT as well as reduced air temperatures during 1790 – 1820 CE as indicated by Stockholm and Central England instrumental time series (Fig. 8). Despite this time period is known to coincide with the Dalton minimum in solar activity (Grove, 1988), it is likely that solar activity played much less role than volcanic activity associated with eruptions of 1809 and 1815 (Wagner and Zorita, 2005). The role of AMOC strength in shaping the LIA cold periods is also somewhat controversial based on marine geological evidence: though the AMOC weakening was proposed as a trigger for the LIA cooling (Bianchi and McCave, 1999), it was argued against (Keigwin and Boyle, 2000) and was not statistically significant in paleoclimate modelling (Van der Schrier and Barkmeijer, 2005). It has even been suggested that Gulf Stream may have experienced warming during this period (e.g. Keigwin and Pickart, 1999), which certainly does not explain low BWT temperatures in our record, as well as low air temperatures over Stockholm and Central England during 1790 – 1820 CE. An explanation for this phenomenon has been proposed by Bjerknes (1965), who postulated, "a decrease in western European winter surface air temperatures to be related almost completely to an anomalous southward advection of cold polar air", a hypothesis later verified by a model study of Van der Schrier and Barkmeijer (2005).Âż

Please also note the supplement to this comment:
https://www.clim-past-discuss.net/cp-2017-160/cp-2017-160-AC1-supplement.pdf
* * *
[Figure]

**Supplement:**

[revised manuscript text omitted]

Borttaget: h

Borttaget: based on
Borttaget: observations
Borttaget: performed in the
Borttaget: a
Borttaget: mass
Borttaget: s
Borttaget: is
Borttaget:
Borttaget: , which often does not occurs in until the winter of the following year to come
Borttaget: spring, summer and autumn seasons, all being
Borttaget: the
Borttaget: deep-water
Borttaget: the
Borttaget: f

Borttaget: trigger

Borttaget: usually dropping below 2 ml O$_2$ l$^{-1}$

2C) but due to the low observation frequency and duration of these events are not well documented. Since 1979, multiple episodes of more frequent severe hypoxia lasting for at least 3 months have been observed. These events occurred in 1979/80, 1983/84, 1987/88, 1988/89, 1990/91, 1994/95, 1996–1998, 2008, 2014/2015, 2016 (e.g. Filipsson and Nordberg 2004a; Polovodova Asteman and Nordberg 2013; SMHI SHARK-database, 2017; Nordberg, unpubl. data).

5 The severe hypoxia makes the fjord basin hostile for large burrowing organisms but allows benthic meiofaunas to thrive. This lowers sediment bioturbation and results in well-preserved environmental sediment archive. The fjord basin has high sediment accumulation rates, which provide a high temporal resolution corresponding to 1-6 years per 1-cm thick sediment sample. Finally, the fjord sediment archive is characterized by the diverse and abundant foraminiferal faunas and dinoflagellate cysts, which have already provided some insights in climate evolution and associated environmental changes

10 on the Swedish west coast during the last two millennia (Filipsson & Nordberg, 2004a; Harland et al., 2006; Nordberg et al. 2009; Filipsson and Nordberg 2010; Polovodova et al., 2011; Harland et al., 2013; Polovodova Asteman & Nordberg, 2013; Polovodova Asteman et al., 2013).

**3 Material and Methods**

This study is based on a composite record of two sediment cores: GA113-2Aa and 9004, which were both collected at 116 m

15 water depth at the same site in the deepest Gullmar Fjord basin (58°17.570' N, 11°23.060' E) (Fig. 1), for which the long-term hydrographic observations are available (Fig. 2A-C). The core 9004 (731-cm long) was taken with a gravity corer (Ø=7.6 cm) onboard *R/V Svanic* in July 1990. The core GA113-2Aa (60-cm long) with an intact sediment-bottom water interface was recovered by using a Gemini corer (Ø=8 cm) in June 1999 from the *R/V Skagerak*. In the laboratory both cores were split in two halves and sectioned in 1-cm intervals. One half was used for bulk sediment geochemistry (TC, TN and

20 C/N ratio), stable oxygen and carbon isotopes, dinoflagellate cysts- and benthic foraminiferal faunal analyses. Another half was stored as an archive at the Department of Geosciences, University of Gothenburg. The TC and stable carbon isotope data from both cores are published in Filipsson and Nordberg (2010), dinoflagellate cysts data are discussed in Harland et al. (2006, 2013), while C/N and foraminiferal assemblage data are presented in Filipsson & Nordberg (2004a), Polovodova et al. (2011) and Polovodova Asteman et al. (2013). We also present data from the gravity core G113-091, collected at the

25 same location as GA113-2Aa & 9004 onboard *R/V Skagerak* in September 2009, and used herein only (similar to our previous study) to create a composite age model for the cores GA113-2Aa and 9004 (Polovodova Asteman et al., 2013; see below).

In addition to the above-mentioned cores, we also use six surface samples (0-1cm) collected at five stations in the Skagerrak (OS4, OS6, OS14, 9202 and 9205) and one station in the Gullmar Fjord (G113-091a: the same location as for

30 GA113-2Aa & 9004) in 1992-93 and 2009, respectively (Fig.1B, C; Table 1). All surface samples were stained by rose Bengal to distinguish individuals presumably living at the moment of sampling from the empty foraminiferal shells.

**Borttaget:** aboard

**Borttaget:** of

**Borttaget:** aboard

**Borttaget:** of

**Formaterat:** Teckensnitt:Inte Fet, Kursiv, Teckenfärg: Auto

[revised manuscript text omitted]

**Borttaget:** correlation

**Borttaget:**

**Borttaget:** Orange rectangle shows recent BWT range (3.0-8.1°C) for instrumental observations performed during 1961-1990.

**Borttaget:** (

**Borttaget:** )

**Figure 7:** Comparison of the winter bottom water temperatures (BWT) reconstructed from Gullmar Fjord record to instrumental basin water temperatures measured in the deepest fjord basin: the annual mean (a), mean for May-August (b) and mean for January-March (c).

**Figure 8:** Comparison of reconstructed winter bottom water temperatures (BWT) from Gullmar Fjord to meteorological observations of winter air temperatures recorded for Stockholm (stippled line) and the Central England (solid line without symbols).

**Supplementary Figure 1:** Scatter plot of stable carbon isotopes ($\delta^{13}$C) data from the composite G113-2Aa – 9004 record (Filipsson and Nordberg, 2010) plotted against the oxygen isotope data presented herein. Note absence of correlation between the two, ruling out the possibility that the changes in $\delta^{18}$O are due to changes in water masses.

**Table captions:**

**Table 1:** Stations with collected sediment core tops and $\delta^{18}$O analyzed on living (rose Bengal stained) *Cassidulina laevigata*.

**Table 2.** AMS $^{14}$C dates obtained for the gravity core 9004 and calibrated calendar ages. All dates presented in Filipsson and Nordberg (2010) and Polovodova Asteman et al. (2013) were re-calibrated using Calib 7.10 (Stuiver *et al*. 2017), the Marine13 calibration dataset (Reimer et al, 2013), and $\Delta R = 100 \pm 50$. Asterisks (*) show dates not used in the final age model due to age reversals.

---

## Author Response (AR1)

**Author's response to reviewer 1**

We sincerely thank reviewer 1, Antoon Kuijpers, for his useful and insightful comments and an advice on highly relevant references, which we missed to include. Below we respond to the comments point by point:

Comment 1:
Comment/ discuss general results in context of (non-cited) highly relevant reference Luterbacher et al. 2016 Env.Res.Lett. 11.

Response:
We first had troubles tracking down the suggested reference, until we have found out that it must be a paper by Orth et al (2016), including Luterbacher as a co-author, since it appears to be the only paper in Env.Res.Lett. 11, which, indeed, turned out to be highly relevant for our discussion. Now we included this into the discussion and added the following section under the "LIA milder episode":
*«Indeed, several studies report an exceptional multi-month drought and long-lasting warm conditions in Europe associated with year 1540 (Casty et al., 2005; Pauling et al., 2006; Wetter et al, 2014), which given our age model uncertainty for the time interval 1538-1664 BCE (±40 yr, see Table 2) may well fall within the warm period identified for the LIA from our BWT record. A warming around 1540 is also seen in winter temperature reconstruction for Stockholm ports and harbours based on historical records of sea ice (Leijonhufvud et al., 2009). The model-based reconstruction by Orth et al (2016) suggests that the European temperatures of 1540 exceeded those of the summer 2003, which was likely the warmest for centuries (e.g. Luterbacher et al, 2016). This is, however, difficult to deduce based on our data, since the fjord BWT record only stretches until ~1996.»*

Comment 2:
Start of LIA : refer to Stuiver et al. 1995, Quat Res 44

Response:
The reference has been included.

Comment 3:
Multi-decadal variability lacking reference to possible link to Atlantic Multidecadal Oscillation (AMO). Within this context interesting to discuss results shown in Fig. 7 with peaking BWT values prior to 1920 coinciding with cold AMO / low N Atlantic sea surface salinities (Reverdin et al. 1994 Progr.Ocean. 33; Reverdin 2010, Journ Clim 23), in following period until ca 1960 BWT at a lower level (during warm AMO), after which again peaking (e.g. at time of 'Great Salinity Anomaly', early 1970's).

Response:
We added a reference to AMO (Enfield et al, 2001) and its link to the multidecadal climate variability (through AMOC) into the introduction. We also added a discussion around high fjord BWT at times of cold AMO and reduced salinities in the North Atlantic:
*"Our record also shows higher BWT prior to the 1920s (Fig. 8), which coincides with the cold AMO (low SSTs) and low sea surface salinities in the North Atlantic and Subpolar Gyre (Reverdin et al. 1994; Reverdin, 2010), while in the following period until ~1960, the reconstructed BWT remains at a lower level (during the warm AMO, i.e. high North Atlantic SSTs), after which it peaks again at time of "Great Salinity Anomaly" during the late 1970s and late 1980s (Dickson et al., 1988; Belkin et al., 1998). It remains intriguing, though, that at both occasions (prior to the 1920s and during the 1970s/1980s) of the reduced salinities and low SSTs in the North Atlantic, our record is characterized by high temperatures of the fjord deep water, which is consistent with increasing air temperatures in instrumental datasets from Stockholm and Central England (Fig. 8). The low surface salinities of the Great Salinity Anomaly were likely driven by an increased freshwater/sea ice export from the Arctic via Fram Strait and Canadian Archipelago (Belkin et al., 1998). The increased freshwater flux into the subpolar North Atlantic, in turn, is suggested to increase salinity of the North Atlantic Current, which may reduce its predicted weakening due to enhanced freshwater fluxes and will help to restart a stronger AMOC (Hátún et al., 2005; Thornalley et al., 2009). A stronger North Atlantic Current would in turn result in an increased heat transport during winter to the Eastern North Atlantic and together with other external forcing factors (e.g. changes in NAO, volcanism, and solar activity) would contribute to the warming observed in the fjord BWT record during the early 20th century. One of those factors, the positive NAO mode, which prevailed since the 1970s/1980s (Hurrell, 1995;*

http://www.cpc.ncep.noaa.gov/products/precip/CWlink/pna/season.JFM.nao.gif), extracts heat from the subpolar North Atlantic through increased westerlies over that region, decreases SSTs, enhances convection, increases ocean density (Delworth et al., 2016; Delworth and Zeng, 2016) and results in milder winter conditions over the north-western Europe, thus counteracting effects of the AMOC weakening, which has been suggested for the 20$^{th}$ century based on modeling data and proxy records (Caesar et al., 2018; Thornalley et al., 2018). Also, located within a coastal region, the Gullmar Fjord is more susceptible to wind-forced temperature changes, which follow the variability of the NAO index and drive coastal upwelling and downwelling in the fjord (Björk and Nordberg, 2003). According to Jansen et al. (2007), the late 20$^{th}$ century warming as demonstrated by many proxy records from the NE Atlantic (see discussion above), is unlikely to be explained by the external forcing factors and is probably linked to the anthropogenic drivers such as greenhouse gas emissions and aerosols (Booth et al, 2012), which both significantly increased since ~1970s (Masson-Delmotte, 2013)."
."

Comment 4:
Fig. 8: Discuss Dalton Minimum (AD 1790 - 1820) with general low T, both Tair and BWT coinciding with Gulf Stream warming (see previous remark !) , ref Van der Schrier and Barkmeijer 2005, Clim Dyn. 24

Response:
We agree with the reviewer and have added a following section into the LIA-discussion:

"Another interesting feature of the LIA climate variability is associated with consistently low fjord BWT as well as reduced air temperatures during 1790 – 1820 CE as indicated by Stockholm and Central England instrumental time series (Fig. 8). Despite this time period is known to coincide with the Dalton minimum in solar activity (Grove, 1988), it is likely that solar activity played much less role than volcanic activity associated with eruptions of 1809 and 1815 (Wagner and Zorita, 2005). The role of AMOC strength in shaping the LIA cold periods is also somewhat controversial based on marine geological evidence: though the AMOC weakening was proposed as a trigger for the LIA cooling (Bianchi and McCave, 1999), it was argued against (Keigwin and Boyle, 2000) and was not statistically significant in paleoclimate modelling (Van der Schrier and Barkmeijer, 2005). It has even been suggested that Gulf Stream may have experienced warming during this period (e.g. Keigwin and Pickart, 1999), which certainly does not explain low BWT temperatures in our record, as well as low air temperatures over Stockholm and Central England during 1790 – 1820 CE. An explanation for this phenomenon has been proposed by Bjerknes (1965), who postulated, "a decrease in western European winter surface air temperatures to be related almost completely to an anomalous southward advection of cold polar air", a hypothesis later verified by a model study of Van der Schrier and Barkmeijer (2005).»

**Author's response to reviewer 2**

We sincerely thank anonymous reviewer 2 for insightful comments and valuable suggestions on how to improve the manuscript. Below we respond to each of the comments.

Major comment 1:
**Relevance:** The authors need to explain better why the study is important. It is mentioned in the introduction that only few high-resolution records of late Holocene conditions exist from the eastern North Atlantic region. But records also exist from other regions, both Iceland and the western North Atlantic and the Labrador Sea region. Why is the Eastern North Atlantic region important? Please add a short explanation, what is special/different about this region compared to other areas. How can this study improve our general understanding of the late Holocene climate of the North Atlantic and which mechanisms control climate and ocean variability?

Response:
We are aware of the existing records from other regions, such as Iceland, the W North Atlantic and the Labrador Sea (among many others Jiang et al., 2005; Andresen et al, 2012; Seidenkrantz et al., 2012; Perner et al., 2011, 2013, Sicre et al., 2008, 2014 etc). However, in this paper we merely wanted to stay focused on available fjord records from the NE Atlantic, which all share high temporal resolution (i.e. annual to subdecadal) and similar fjordic hydrography allowing calm sedimentation and continuous sediment accumulation with minor dilution by glaciomarine and/or terrigenous component. One of the reasons for a specific focus on the NE Atlantic is due to a geographical location of the majority of the NH temperate silled fjords, which are simply less frequent on the western side of the Atlantic. In addition, we believe that since the North Atlantic Current (NAC), and it's northward extension in a form of the Norwegian Atlantic Current, is one of the branches carrying a major part of the volume flux (and hence heat and salt) into the Nordic Seas *(*Hansen and Østerhus, 2000) and its ameliorating effect on NW European climate, more high-resolving records from sites influenced by the NAC and having long-term instrumental observations are needed. This is especially important in view that future predictions warn about NAC weakening (through AMOC) due to greenhouse forcing and, also, given the AMOC close connection to other mechanisms and phenomena controlling climate and ocean variability not only locally (for NW Europe) but also regionally and through teleconnections (e.g. NAO, AMO, ENSO).

We added the following section into the introduction following the reviewer's suggestion:
"*The North Atlantic region plays in this respect a paramount role for climate variability and global carbon budget by modulating the Atlantic Meridional Overturning Circulation (AMOC) (e.g. Eiríksson et al., 2006; Lund et al., 2006; Park and Latif, 2008; Trouet et al., 2009). The upper northern limb of the AMOC, the North Atlantic Current, delivers heat, salt and nutrients from tropics to the mid- and high latitudes and carries a major part of the volume flux into the Nordic Seas (Hansen and Østerhus, 2000). The AMOC is thought to be linked to the sea surface temperature variability of the Atlantic multidecadal oscillation (AMO; Enfield et al., 2001) and is connected to decadal variability of the North Atlantic Oscillation (NAO), where the NAO index is defined as the normalized sea level pressure difference between the Icelandic Low and the Azores High (Hurrel et al., 1995). The AMOC also contributes to a multidecadal modulation of El Niño-Southern Oscillation (ENSO) (Ortega et al., 2012 and references therein). Finally, variability of ocean temperature in high latitude North Atlantic and Nordic Seas are reflected in NW European climate and in winter Arctic sea ice extent (Årthun et al., 2017). Model projections predict AMOC slowdown in response to future warming and enhanced Arctic freshwater fluxes (e.g. Schmittner et al., 2005; Ortega et al., 2012) with potential detrimental impacts on the climate, the ecosystems and the economy of many European countries (e.g. Kuhlbrodt et al., 2009; Jackson et al., 2015). Hence, high-resolution paleoceanographic records, which preferably overlap with instrumental observations and historical data, are needed from the eastern North Atlantic region in order to document climate variability related to physical properties of the North Atlantic Current and AMOC strength.*"

Major comment 2:
**Bottom Water Temperatures:** Page 3, line 28-31. It is stated that the water exchange only occurs during winter. Does any change in salinity or temperature conditions of the bottom waters occur during spring/summer?
Explain more clearly whether the Bottom Water Temperatures actually represent winter conditions (mention this also in the abstract). As this is a central part of the work, it needs to be explained very clearly.
Seasons used in the bottom-water temperature reconstruction (p9): Traditionally the winter season is described through the months DJF, but here the period JFM is used. Why? Is there a local environmental reason for this, purely due to available data, or…Similarly an explanation should be given for the use of May-August as the summer period, but this is normally JJA. It is not directly stated in paragraph 4.3 that these periods correspond to "winter" (JFM) and "summer (MJJA) but in the following discussion (paragraph 5) winter temperatures are mentioned, so I assume that this is the case? However, it needs to be stated clearly and explained properly.

Response:
We tried to be more specific and to clarify the "winter temperature signal" by modifying the corresponding Study area section accordingly:
*" The deep water temperatures vary between the years depending on the temperature of the inflowing water mass but remain stable seasonally (Fig. 2D). The deep-water salinities seasonally do not vary much from the average value of 34.5 (Fig. 2B). The stratification of the water column is strengthened during the summer by the development of a strong thermocline, which impedes deep-water exchange. The deep-water exchange of the fjord basin water takes place once a year during winter, mostly between January and March, based on long-term instrumental observations performed in the fjord (Arneborg et al., 2004). Due to a presence of a sill isolating the fjord deep-water mass from the adjacent seas and the large basin volume, the winter temperature and salinity of the inflowing North Sea/Skagerrak water, are "annually preserved" in the fjord basin until the next deep-water turnover, which does not occur until the winter of the year to come (Arneborg et al., 2004). This results in a bottom water environment characterised by the winter temperatures. The benthic foraminifers reproduce and grow in the fjord during the spring and summer (Gustafsson and Nordberg 2001), thus incorporating this annually preserved winter temperature signal of the ambient deep-water into their shells. This results in a stable oxygen isotope signal mainly reflecting winter temperatures of the North Sea surface water and the Skagerrak intermediate water."*

As regarding the choice of JFM for winter months instead of the most commonly used DJF, this is due the deep-water turnover timing (Jan-March) described above. For example this can be seen clearly in Fig. 2 (C-E), where the bottom water exchange occurred in March 1993 (green rectangle). In addition, March is a month included in calculation of the winter NAO index (Hurrell, 1995), which has a documented effect on the deep-water exchange in the fjord (Björk and Nordberg, 2003). In contrast, it is very unlikely for bottom water renewal to occur in December, based on instrumental time series available for the fjord deep basin.

Similarly, May-August, are the months associated with foraminiferal growth in the fjord (Gustafsson & Nordberg, 2001), and hence we use those months when plotting instrumental observations for "summer season", instead of commonly used JJA. We added this information in section 4.3.

Major comment 3:
**General interpretation and potential link to the NAO:**
3a) The section on the influence of the North Atlantic Oscillation (NAO) on the Gullmar Fjord (p 11, lines 1.6) should be moved to the introduction, with reference also to modern data from NE Europe/NE Atlantic. No reference to NAO during past climate periods should be mentioned as fact before this is discussed in the following paragraphs.

3b) The potential role of the NAO is discussed for the MCA and LIA. But what about the RWP and the DACP? Several studies have indicated that climate during these periods may also be linked to the NAO, and the manuscript would benefit form a more in-depth discussion – and reference to a

wider range of previously published studies. It is also noteworthy that the authors only refer to work that shows comparable conditions as seen in Gullmar Fjord, omitting any other studies. The authors should also look towards studies on the Late Holocene from further afield, e.g. Portugal, East Greenland, the Labrador Sea.

Response:
3a) This has been done. We added the general info regarding the NAO into the introduction (see response to major comment 1) and also inserted the following text into the Study area section:
*"The deep-water exchange in the fjord is driven by wind forcing, and largely depends on wind direction and wind strength (Björk and Nordberg, 2003). The latter two properties, in turn, are governed by the NAO, which is the dominant mode of climate variability in the region during the winter. In Gullmar fjord, the higher frequency and duration of NE winds, common during the negative NAO index periods, result in Ekman transport of surface water from the coast and facilitate coastal upwelling, which causes the deep-water exchange (Björk and Nordberg, 2003). In contrast, a positive NAO index causes stronger westerly winds, which prevent the deep-water renewals to occur. From the late 1970s the NAO has been in its prolonged positive phase and is believed to trigger severe seasonal hypoxia in the deep fjord basin (Nordberg et al., 2000; Björk and Nordberg, 2003; Filipsson and Nordberg, 2004).*

3b) We added the potential role of the NAO for the RWP and DACP in the discussion:
For RWP: *"Other studies report an increased contribution of the Atlantic water to the East Greenland shelf, a reduced sea ice concentration and an increased export of fresh water from the Arctic with the East Greenland Current during the RWP, which are thought to be linked to a shift from negative to positive NAO after ~500 BCE/0 CE and changes in the AMO regime (e.g. Perner et al., 2015 and references therein; Kolling et al., 2017)".*

For DACP we also added references to a negative NAO mode as suggested by e.g. Orme et al, 2015 and Helama et al 2017.

Major comment 4:
**Hypoxia:**
4a) On P. 7, line 5 and again Page 10, line 17-18 it is mentioned that C. laevigata has become a rare species in the Gullmar Fjord since 1990. One page 7 no explanation is given, page 10 the phenomenon is explained through hypoxia. However on page 15 a discussion is raised, whether it is due to hyposix and if yes, why. The discussion is certainly relevant but the fact that first a statement is made and later a discussion is raised, makes it confusing and somewhat messy. I would suggest just to refer to "see†discussion" instead of jumping the gun on p7 and 10. Also¨ the discussion on p 15 does not really fit well to the remaining text, and a solution may be to move this hypoxia discussion to its own, separate paragraph.

4b)With respect to this discussion, the authors basically explain the hypoxia as due to climate change. However, what about the increased nutrient supply seen due to more intensified farming seen in the general region, may this also play a role? Please discuss.

Response:
4a) We referred to discussion on p.7 and 10 regarding mentioning hypoxia and absence of C. laevigata, following reviewer's suggestion. We also added a separated subsection "Environmental conditions explaining absence or rare occurrence of C. lavigata in the record" to separate the discussion about hypoxia.

4b)We added the following sentence into the discussion: *"To a large extent, the oxygen status of fjords and estuaries on the Swedish west coast, is controlled by climate (e.g. Nordberg et al., 2000; Filipsson and Nordberg 2004a, b), but the late Holocene changes in land use and organic enrichment in the fjord are also suggested to play a role (Filipsson and Nordberg, 2010)."*

**Conclusions:**

The paragraph should be expanded with a synopsis on the discussion on the processes driving the climate change.

Response:

We included the following sentences into the section:

*"Those warming (cooling) intervals during the last 2500 years were likely caused by the strengthening (weakening) of the AMOC linked to changes in atmospheric and oceanic forcing, such as the NAO, the AMO and, also perhaps, the ENSO, as suggested by other studies. In addition, changes in solar activity, volcanic forcing and, more recently, the anthropogenic greenhouse gas emissions and aerosols have also been important drivers of the observed climate variability during the late Holocene."*

**Minor comments:**

"Foraminiferal species: add author name to the species name the first time a species is mentioned: i.e., *Cassidulina laevigata* d'Orbigny, 1826; *Adercotryma glomerata* (Brady, 1878); *Hyalinea balthica* (Schröter in Gmelin, 1791)." - *This has been corrected and the author's names have been added.*

"P5, line 14; reservoir correction: How many bivalve shells and from how many sites in the Gullmar Fjord is this reservoir correction based on?" – *The shells were taken at four sites in a fjord deep basin (>100 m) and in total 14 samples were analysed for $^{14}C$ reservoir effect. Four replicates were taken from each of 3 stations, while 1 station had only 2 replicates. The average reservoir age based on those 14 analysed samples is 497±30yr (Nordberg & Possnert, unpubl. data).*

"P.9, line 24: add reference for timing of the foraminiferal growth season." – *The reference has been added.*

"P10, line 22-25: add references for the mentioned climatic intervals." – *All references are present in the text above, just before the climate intervals are mentioned, however we also added some new references as suggested by reviewer 1.*

"Page 15: Could the stronger recent warming of the Marlangen Fjord region be due to a more direct link to the northward flow of Atlantic water compared to Gullmar Fjord, which is not in direct contact with the core of the Atlantic water?" – *It is true and we added this argument into discussion.*

**Figures and figure captions:**

"All terms and abbreviations should be explained."– *This has been corrected.*

"Fig 1: explain abbreviations for current names" – *This has been done.*

"Fig 1a: land masses are shown in a very pale gray – it would be easier to see, if landmasses were shown in a slightly darker colour." – *The figure was changed accordingly*.

"Fig. 2: BWT needs to be explained wither in the figures or the figure captions, as it should be possible to understand the figures without reading the main text." – *This has been done.*

"Fig 3A: I cannot distinguish between the upper and lower symbol; please make them more different." – *The symbols have been changed.*

"Fig. 5: explain BWT, RWP, DA, LIA etc in the figure caption. Mark the present BWT range on the figure." – *This has been done.*

"Fig. 6: explain the pink and blue intervals." – *Has been done, as well.*

"Fig. 7: Here "bottom water temperature" is written in full (not giving the abbreviation) – be

consistent." – *This has been corrected.*

"Some additional comments are provided as comments in pdf file of the manuscript (only relevant pages)." – *Those changes were applied to the text accordingly.*

[revised manuscript text omitted]

**Formaterat:** Upphöjd

**Formaterat:** Teckensnitt:10 pt

**Formaterat:** Upphöjd

**Formaterat:** Teckensnitt:Inte Kursiv

**Formaterat:** Teckensnitt:Inte Kursiv

**Formaterat:** Nedsänkt

**Formaterat:** Justerat, Indrag: Vänster: 0 cm, Hängande: 0,75 cm, Avstånd Efter: 0 pt, Radavstånd: 1,5 rader, Ingen numrering, Ingen kontroll av enstaka rader, Justera inte mellanrum mellan latinsk och asiatisk text, Justera inte mellanrum mellan asiatisk text och siffror, Teckensnittsjustering: Auto, Mönster: Inget

Wetter, O., Pfister, C., Werner, J.P., Zorita, E., Wagner, S., Senevirante, S.I., Herget, J., Grünewald, U., Luterbacher, J., Alcoforado, M.-J., Barriendos, M., Bieber, U., Brázdil, R., Burmeister, K.H., Camenish, S. et al.: The year-long unprecedented European heat and drought of 1540—a worst case, Clim. Change 125, 349–63, 2014.

Zicheng, Y. & Ito, E.: Historical solar variability and midcontinent drought. Pages Newsl. 8, 6–7, 2000.

5   Årthun, M., Eldervik, T., Viste, E., Drange, H., Furevik, T., Johnson, H.L., Keenlyside, N.S.: Skillful prediction of northern climate provided by the ocean, Nature Comm. 8, 15875, DOI: 10.1038/ncomms15875, 2017.

**Datasets used in this paper were extracted from:**

SMHI, SHARK database https://www.smhi.se/klimatdata/oceanografi/havsmiljodata/marina-miljoovervakningsdata, accessed 15/03/2017.

10   SMHI: Meteorological observations of air temperatures: https://www.smhi.se/klimatdata, accessed 15/03/2017.

ICES database: http://www.ices.dk/marine-data/, accessed 08/03/2017.

Central England air temperature dataset: http://www.metoffice.gov.uk/, accessed 16/03/2017.

**Figure captions**

**Figure 1:** Map of the study area including location of Gullmar Fjord (GF) and sampling site of Ga113-2Aa & 9004 record (star) within North Atlantic (A) and North Sea – Skagerrak region (B). Locations of other discussed proxy records are shown by white circles, while some of major ocean circulation characteristics mentioned in the text are indicated as: EGC – East Greenland Current, NAC – North Atlantic Current, SPG – Subpolar Gyre, and STG – Subtropical Gyre (A). B: the major

20   regional water masses and currents are shown as follows: AW - Atlantic Water, SJC – South Jutland Current, NJC – North Jutland Current, BC – Baltic Current, NCC – Norwegian Coastal Current. C: an overview of water column stratification in the longitudinal profile of the Gullmar Fjord with indication of salinity (S) and residence times (t) typical for each water layer (Arneborg, 2004).

25   **Figure 2:** Hydrographic measurements from Alsbäck Deep, Gullmar Fjord taken during 1890 – 2000 below 110 m water depth: BWT – bottom water temperature (a), salinity (b) and dissolved oxygen (c). A snapshot of hydrographic changes in BWT (d), salinity (e) and oxygen (f) associated with basin water exchanges between 1992 and 1993 showing annual variability of these parameters.

**Figure 3:** Age model of the studied Ga113-2Aa & 9004 record (A) and comparison of foraminiferal and isotopic data with core G113-091, taken at the same location in 2009, to prove the absence of a gap between GA113-2Aa and 9004 (B), according to Polovodova Asteman et al (2013).

**Figure 4:** Comparison of reconstructed temperatures and δ¹⁸O values measured in stained *C. laevigata* from the core tops collected in Gullmar Fjord (G113-091) and the Skagerrak (OS4, OS6, OS14, 9202, 9205) to hydrographic temperature data (A) and to δ¹⁸O predicted from palaeotemperature equation (B) by McCorkle et al (1997). C: Temperature vs. δ¹⁸Oc – δ¹⁸Ow, together with the paleotemperature equations from Shackleton (1974), Hays and Grossman (1991), Kim and O'Neil (1997), McCorkle et al. (1997), and Bemis et al. (1998).

**Figure 5:** A 2500-year long δ¹⁸O record (A) and reconstructed winter bottom water temperatures, BWT (B) from Gullmar Fjord. Thick lines show 3-point running mean for both curves, and dashed lines indicate A) a long-term average of 2.4‰ for δ¹⁸O record and B) 5.4°C - a mean for instrumental bottom water temperatures registered between 1961 and 1990. Grey shaded areas in BWT indicate a median offset (0.7°C) in instrumental versus reconstructed temperatures obtained for rose Bengal stained *C. laevigata* from the core tops (see Fig. 4A), used herein as an error margin. C: Box and whisker plot showing a range for instrumental BWT observations performed during 1890 – 1999 and measured at more regular intervals from the 1960s, the data is from water depths ≥110 m in the fjord deepest basin (Alsbäck Deep). The middle, the upper and the lower horizontal lines in the box indicate the median, 75 and 25 percentiles, respectively. Abbreviations are as follows: RWP – the Roman Warm Period, DA – the Dark Ages, VA/MCA – the Viking Age/Medieval Climate Anomaly and LIA – the Little Ice Age.

**Figure 6:** Reconstructed bottom water temperatures (BWT) shown as anomaly against the 1961-1990 instrumental mean of 5.4°C from Gullmar Fjord compared against other temperature proxy records: annual northern hemisphere temperatures (Moberg et al., 2005), bottom water temperatures from Malangen Fjord in NW Norway (Hald et al., 2011) and Loch Sunart in Scotland (Cage and Austin, 2010), spring sea surface temperatures from Chesapeake Bay, E North Atlantic Ocean (Cronin et al., 2003), annual temperatures reconstructed for continental Europe (Pages2K, 2013) and the reconstructed NAO record from Trondheim Fjord, W Norway (Faust et al., 2016). Also are shown relative abundances of foraminifer *Cassidulina laevigata* in the fjord with abundance minima and respective gaps in temperature reconstruction linked to the positive NAO index (arrows). For location of these proxy records see Fig. 1A and for abbreviations see text to Fig. 5. Grey shaded areas in Gullmar Fjord BWT anomalies indicate a median offset (0.7°C) in instrumental versus reconstructed temperatures (see Fig. 4A) obtained for rose Bengal stained *C. laevigata* from the core tops, used herein as an error margin. Blue and pink boxes depict a short-lived cooling at ~1250 CE and a warm interval between ~1570 and 1700 CE, both of which are discussed in the text.

**Borttaget:** correlation

**Borttaget:**

**Borttaget:** Orange rectangle shows recent BWT range (3.0-8.1°C) for instrumental observations performed during 1961-1990.

**Borttaget:** (

**Borttaget:** )

**Figure 7:** Comparison of the winter bottom water temperatures (BWT) reconstructed from Gullmar Fjord record to instrumental basin water temperatures measured in the deepest fjord basin: the annual mean (a), mean for May-August (b) and mean for January-March (c).

**Figure 8:** Comparison of reconstructed winter bottom water temperatures (BWT) from Gullmar Fjord to meteorological observations of winter air temperatures recorded for Stockholm (stippled line) and the Central England (solid line without symbols).

**Supplementary Figure 1:** Scatter plot of stable carbon isotopes ($\delta^{13}C$) data from the composite G113-2Aa – 9004 record (Filipsson and Nordberg, 2010) plotted against the oxygen isotope data presented herein. Note absence of correlation between the two, ruling out the possibility that the changes in $\delta^{18}O$ are due to changes in water masses.

**Table captions:**

**Table 1:** Stations with collected sediment core tops and $\delta^{18}O$ analyzed on living (rose Bengal stained) *Cassidulina laevigata*.

**Table 2.** AMS $^{14}C$ dates obtained for the gravity core 9004 and calibrated calendar ages. All dates presented in Filipsson and Nordberg (2010) and Polovodova Asteman et al. (2013) were re-calibrated using Calib 7.10 (Stuiver *et al*. 2017), the Marine13 calibration dataset (Reimer et al, 2013), and $\Delta R = 100 \pm 50$. Asterisks (*) show dates not used in the final age model due to age reversals.

Figure 1

[Figure]

**Figure 2**

[Figure]

Figure 3

[Figure]

Figure 4

[Figure]

Figure 5

[Figure]

Figure 6

[Figure]

Figure 7

[Figure]

Figure 8

[Figure]

Supplementary Fig. 1

[Figure]

**Table 1**: Stations with collected sediment core tops and $\delta^{18}O$ analyzed on living (rose Bengal stained) *Cassidulina laevigata*.

| Station | Latitude N | Longitude E | Water depth, m | Sampling date | $\delta^{18}O$, ‰ |
|---------|-----------|-------------|----------------|---------------|---------|
| 9202 | 57°56.2' | 9°27.3' | 177 | 1992-08-04 | 2.49 |
| 9202 | 57°56.2' | 9°27.3' | 177 | 1992-08-04 | 2.44 |
| 9205 | 57°58.4' | 9°24.0' | 294 | 1992-08-06 | 2.40 |
| OS14 | 58°06.06' | 10°58.27' | 135 | 1993-05-09 | 2.58 |
| OS4 | 58°18.54' | 8°54.99' | 325 | 1993-05-04 | 2.48 |
| OS6 | 58°21.58' | 8°51.01' | 177 | 1992-08-04 | 2.43 |
| G113-091 | 58°17.570' | 11°23.060' | 116 | 2009-09-01 | 2.76 |

**Table 2.** AMS $^{14}$C dates obtained for the gravity core 9004 and calibrated calendar ages. All dates presented in Filipsson and Nordberg (2010) and Polovodova Asteman et al. (2013) were re-calibrated using Calib 7.10 (Stuiver *et al*. 2017), the Marine13 calibration dataset (Reimer et al, 2013), and ΔR = 100 ± 50. Asterisks (*) show dates not used in the final age model due to age reversals.

| Core | Core depth (cm) | Lab. ID | Dated bivalve species | $^{14}$C age (years BP) | Error (±) | Calibrated age range, ±1σ, ΔR=100±50 (years CE/BCE) | Relative probability | Calibrated age, median probability (years CE) |
|---|---|---|---|---|---|---|---|---|
| 9004 | 98 | Ua-24043 | *Nuculana minuta* | 710* | 35* | 1645-1806* | 1 | 1702* |
| 9004 | 136 | Ua-35966 | *Nuculana pernula* | 675 | 25 | 1675-1813 | 1 | 1750 |
| 9004 | 164 | Ua-23075 | *Yoldiella lenticula* | 800 | 40 | 1538-1664 | 1 | 1599 |
| 9004 | 265 | Ua-35967 | *Nucula* sp. | 1025 | 30 | 1356-1372/1383-1465 | 0.106/0.894 | 1416 |
| 9004 | 312 | Ua-35968 | *Clamys septemradiatus* | 1145 | 25 | 1295-1389 | 1 | 1336 |
| 9004 | 313 | Ua-23000 | *Abra nitida* | 1305* | 45* | 1138-1276* | 1 | 1195* |
| 9004 | 371 | Ua-35969 | *Nucula tenuis* | 1245 | 25 | 1208-1303 | 1 | 1251 |
| 9004 | 492 | Ua-23001 | *Abra nitida* | 1640 | 45 | 776-938 | 1 | 853 |
| 9004 | 564 | Ua-23002 | *Nuculana minuta* | 1925 | 40 | 517-658 | 1 | 576 |
| 9004 | 623 | Ua-23003 | *Thyasira flexuosa* | 2155 | 45 | 246-410 | 1 | 321 |
| 9004 | 705 | Ua-23004 | *Thyasira flexuosa* | 2415 | 45 | 68 BCE-102 CE | 1 | 16 |

---

## Author Response (AR2)

Dear editor,

We have changed the manuscript according to the most relevant comments of the reviewer 2 (please see Author's final response to comments).
However, when it comes to the last comment regarding which processes have played a role in climate variability during different periods (RWP, DACP etc), we find it too speculative since our study is proxy-based and has nothing to do with modelling, by which it would be more appropriate to address such a question. By using proxy data we can only compare our results to other regional and global studies, which have linked climate changes to one or another mechanism. That is why we shaped our conclusions as they appear in their present form in hope that our dataset can be used by modellers in the future to clarify the role of different climate change drivers during the different periods. We hope that you also see this as wise and thoughtful decision and will accept our choice on how we prefer to present our dataset and conclusions in the paper.

We attach the revised manuscript with markups indicating applied changes as supplement to this submission. A detailed response to reviewer's comments is given below.

Best regards,
Kjell Nordberg and the co-authors.

**Author's response to reviewer 2 (Final comments)**

We would like to thank the anonymous reviewer 2 for additional comments on how to improve the manuscript. Below we respond to each of the raised points in details.

Minor comment 1:
**Introduction:** The authors seems to have misunderstood my comment on asking for a better explanation on the significance of the study site. The answered with an extended explanation to the significance of AMOC. This is a good addition, but it would have been nice to get a few words also on why NW European sites are important and why the Gullmarfjorden is especially good. With the addition made this is no longer essential, but would improve the ms.

Response:
There was no misunderstanding, we intentionally expanded the section explaining role of the AMOC and especially the role of its NE limb (North Atlantic Current) delivering heat to Northern Europe, thus underlining the role of NE Atlantic sites in reconstructing climate variability linked to the variability of the N Atlantic Current (and hence AMOC) during the last 2 millennia. Also we write on multiple occasions that Gullmar Fjord provides a unique archive for climate reconstructions not only due to its location in NE Atlantic region but also due to its high temporal resolution caused by high sediment accumulation rates and a winter signal, which unlike to many other locations, here is recorded in foram shells. Therefore we do not think (and as reviewer himself/herself admits that it is no longer essential) that manuscript would benefit from narratively repeating this information several times in the text.

We added though a short statement into the introduction on p.3, lines 20-22:
"*Among advantages of the presented record are its high temporal (annual to sub-decadal) resolution and a winter temperature signal, which is unlike to most other proxies is recorded in fjord foraminiferal shells due to specific hydrographic conditions.*"

Minor comment 2:

**§5.1 The RWP:** Remember author name for "Cassidulina neoteretis" and check that author names have been added to all species (the first time).

Response: This has been corrected to *Cassidulina neoteretis* Seidenkrantz 1995 on p. 13, line 3.

Minor comment 3:
**§5.2 The DACP:** During the DACP there was warming of subsurface waters off West Greenland (see e.g. Seidenkrantz et al. 2007, Holocene), ascribed to a stronger Atlantic component of the West Greenland Current and a negative NAO.

Response: We added a following sentence to p. 13, lines 25-26: "*Seidenkrantz et al. (2007) also report a warming of subsurface waters off West Greenland during the DACP attributed to a stronger Atlantic component of the West Greenland Current and a negative NAO.*"

Minor comment 4:
**Page 17 lower part**: please add more detailed explanation to the link between cold AMO phase and cold bottom water – this link may not be logical to everyone.

Response:
The must be a misunderstanding here. As it was emphasized in the report by reviewer 1, on both occasions of the cold AMO (prior to the 1920s and during the 1970s/1980s) we observe WARM bottom water in the fjord (not COLD) and this phenomenon is already explained explicitly on p. 17. Investigating the role of AMO in the variability seen in our record was never a major goal of this paper, since we have a bottom water temperature record, and not a SST record, which would be more directly related to AMO (=SST) variability. Even though the Gullmar fjord deep water originates from the North Sea surface water, we know very little about processes modifying properties of this water mass on its way to the fjord. Hence, to avoid unnecessary speculations we would prefer to not extend this section further.

Minor comment 5:
**Conclusions:** final 4 lines: can it be concluded which of the mechanisms are most important and under which conditions? Right now, it is rather imprecise.

Response:
This study is based on proxy data, and not on modeling results. By modeling it would be possible to adjust variables and discuss more in depth about different mechanisms which were important and when. By using proxy data, compared to other studies, one can only speculate about one or other mechanism to play a role. Therefore we chose to present our most important results and not to speculate too much about different mechanisms, not least because our dataset ends in 1996 after whcih species *Cassidulina laevigata* became extremely rare in the fjord, as a result of the establishment of severe seasonal hypoxic conditions in the bottom water.

---

## Author Response (AR3)

Dear editor,

Please find attached the revised and final version (without markups) of our manuscript.
We have now addressed the following minor points as follows:

Comment 1. Please put the doi for the Pangea data, this will make it easier for anyone to find your data.

Response: We have contacted PANGAEA but as it seems due to increased amounts of datasets being currently submitted the issue of doi: number can take some time. To solve this we can either refer to the published paper and its doi, so they both are connected in PANGAEA depository and are easy to find. Another way would be to contact the CP journal again when the doi is issued so it can be inserted into the publication at later stage. Please let us know if we should do the latter.

Comment 2. I would suggest, in the final version, to advise readers to use the conclusions with the due caution since your dataset does not go beyond the year 1996 due to lack of material and, hence, does not cover the most recent years of warming. This might fit in either the discussion or conclusion section.

Response: We have now added the following lines into the MS discussion (please see p18, lines 23-27):
"When studying the Gullmar Fjord bottom water temperature record for the last 2500 years, it is interesting to note that the most recent warming of the 20th century (presented herein until 1996) does not stand out but appears to be comparable to both the Roman Warm Period and the Medieval Climate Anomaly. This observation has, however, to be used with caution since our dataset does not go beyond year 1996 due to a lack of material and, hence, does not cover the most recent part of the 20th century warming, widely accepted as triggered by growing anthropogenic emissions."

We also rephrased a bit the last part of the conclusions to the following:
"The record also picks up the contemporary warming of 1930s and the 1990s. When studying the Gullmar Fjord bottom water temperature record for the last 2500 years, it is interesting to note that the warming of the 20th century (presented herein until 1996) is comparable to both the Roman Warm Period and the Medieval Climate Anomaly.

Comment 3. Could you also explicitly refer to other studies showing the same pattern about the recent warming seen in their datasets when bringing up the contemporary warming? This could be easily inserted in the discussion.

Response: We have now added the following lines into the MS discussion (please see p16, line 32 – p17, lines 1-5):
"Gullmar Fjord temperature record shows that when considering a 3-point running mean temperature variability, the most recent warming does not stand out in comparison to the RWP and the MCA, as it has been also previously demonstrated by other studies such as e.g. a tree ring-based summer temperature record from central Scandinavia (Linderholm and Gunnarson, 2005), the Scottish loch data (Cage and Austin, 2010), the North Atlantic SST composite (Cunningham et al., 2013), and a 2000-year temperature record for continental Europe (PAGES2K, 2013)."

We sincerely hope that you find our response satisfactory and can accept the revised paper for further publication in the Climate of the Past.

Best wishes
Kjell Nordberg and co-authors